# Depth Without the Magic: Inductive Bias of Natural Gradient Descent

## Abstract

In gradient descent, changing how we parametrize the model can lead to drastically different optimization trajectories, giving rise a surprising range of meaningful inductive biases: identifying sparse classifiers or reconstructing low-rank matrices without explicit regularization. This implicit regularization has been hypothesised to be a contributing factor to good generalization in deep learning. However, natural gradient descent is approximately invariant to reparameterization, it always follows the same trajectory and finds the same optimum. The question naturally arises: What happens if we eliminate the role of parameterization, which solution will be found, what new properties occur? We characterize the behaviour of natural gradient flow in deep linear networks for separable classification under logistic loss and deep matrix factorization. Some of our findings extend to nonlinear neural networks with sufficient but finite over-parametrization. We demonstrate that there exist learning problems where natural gradient descent fails to generalize, while gradient descent with the right architecture peforms well.

## 1 Introduction

There is plenty of empirical evidence that the choice of network architecture is an important determinant of the success of deep learning (He et al., 2015; Vaswani et al., 2017). The empirical observations are now supported by theoretical work into the role that parameter-to-hypothesis mapping plays in determining inductive biases of gradient-based learning. Unregularized gradient descent can efficiently find low-rank solutions in matrix completion problems (Arora et al., 2019), sparse solutions in separable classification (Gunasekar et al., 2018) or compressed sensing (Vaškevičius et al., 2019). Valle-Pérez et al. (2018) studied deep neural networks and found evidence that the parameter-hypothesis mapping[1] is biased towards simpler functions as measured by Kolmogorov complexity. Taken together, these observations and findings have lead the community to hypothesize that

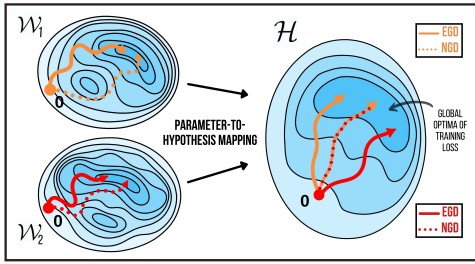

Figure 1: Illustration of parametrization-dependence of EGD and independence of NGD. Consider two parameter spaces ($\mathcal{W}_1$, $\mathcal{W}_2$) and two optimization trajectories in each: one EGD, one NGD. If we map these into the hypothesis space ($\mathcal{H}$) then EGD finds different optima, but NGD finds the same.

*The parameter-to-hypothesis mapping influences the inductive biases of gradient-based learning and may play an important role in generalization.*

In parallel to improving architectures, considerable research was done to improve optimization algorithms for deep learning, with a focus on faster convergence and robustness to hyperparameters. Among the most advanced optimization methods are natural gradient descent (NGD) techniques. An intuitive motivation for NGD is that it improves convergence by implicitly lifting the problem from parameter-space, where the loss is non-convex and poorly behaved to the Riemannian manifold

---

[1]The mapping between the parameter space and the set of hypotheses as seen on Figure 1

of hypotheses, where the loss is better behaved. From the perspective of inductive biases, the most interesting aspect of NGD is its approximate invariance to reparametrization.

*Natural gradient descent eliminates the effect of parameter-to-hypothesis mapping.*

These two observations invite questions about the nature of inductive biases in NGD as well as the role of parametrization-dependence in generalization. The first, practical, implication is as follows: if the parameter-to-hypothesis mapping really does play an important role in generalization, then eliminating its influence on the optimization path may be undesirable, and consequently the pursuit of implementing exact NGD in deep architectures may be counterproductive. Secondly, studying the behaviour of NGD in various models and tasks may give us new insights about the importance of parametrization, and could perhaps offer a way to experimentally or theoretically test hypotheses.

In this paper we study the inductive bias of natural gradient descent in deep linear models. These models are particularly suited for our analysis because (a) efficient algorithms exist to calculate exact natural gradients which is otherwise computationally intractable and (b) the inductive biases of Euclidean gradient descent (EGD) in these models have been thoroughly studied and understood.

We make the following contributions:

- In linear classification, we show that NGF is invariant under invertible transformations of data (Theorems 1&2) and as a consequence it cannot recover the $\ell_p$ large margin solutions that EGD tends to converge to.
- We further show that (in case of separable classification) when the number of parameters exceeds the number of datapoints, NGF interpolates training labels in a way similar to ordinary least squares or ridgeless regression (Theorems 3&4).
- We demonstrate experimentally that there exist learning problems where NGD can not reach good generalization performance, while EGD with the right architecture can succeed.
- To perform experiments, we extended the work of Bernacchia et al. (2018) to derive efficient and numerically stable algorithms for calculating exact natural gradients in diagonal networks (Gunasekar et al., 2018) and deep matrix factorization (Arora et al., 2019).

Before stating our main theoretical and experimental results we review some relevant background on parametrization-dependent implicit regularization and natural gradients.

## 2 BACKGROUND

### 2.1 SEPARABLE CLASSIFICATION WITH DEEP LINEAR MODELS

In this article we consider binary classification datasets $\{(\mathbf{x}_n, y_n), n = 1, \ldots, N\}$ separable by a homogeneous linear classifier with a positive margin ( i.e. $\exists \boldsymbol{\beta}^*$ s.t. $y_n \mathbf{x}_n^\top \boldsymbol{\beta}^* \geq 1 \ \forall n$). (We use the notation $X = (\mathbf{x}_1 \cdots \mathbf{x}_N)^\top$). In such situation $\boldsymbol{\beta}^*$ is not unique and there may be many separating hyperplanes which all achieve 0 training loss - it is up to the inductive biases of the learning algorithm to select one. Soudry et al. (2017) studied the dynamics of unregularized Euclidean gradient descent on logistic loss and found that the iterate $\boldsymbol{\beta}(t)$ converges to the well-known $\ell_2$ large margin classifier in direction, that is

$$\lim_{t \to \infty} \frac{\boldsymbol{\beta}(t)}{|\boldsymbol{\beta}(t)|} = \frac{\boldsymbol{\beta}^*_{\ell_2}}{|\boldsymbol{\beta}^*_{\ell_2}|} \text{ where } \boldsymbol{\beta}^*_{\ell_2} = \arg\min_{\boldsymbol{\beta} \in \mathbb{R}^D} ||\boldsymbol{\beta}||_2 \text{ s.t. } y_n \mathbf{x}_n^\top \boldsymbol{\beta} \geq 1 \quad \forall n.$$

Importantly, Gunasekar et al. (2018) later showed that this behaviour changes if the gradient descent is performed on a different parametrization. In this paper we will focus on $L$-layer linear diagonal networks (Gunasekar et al., 2018), where $\boldsymbol{\beta} = \mathbf{w_1} \odot \mathbf{w_2} \odot \ldots \odot \mathbf{w_L}$, using $\odot$ to denote elementwise product. When we adjust parameters $\mathbf{w_1}, \ldots, \mathbf{w_L}$ through Euclidean gradient descent, $\boldsymbol{\beta}(t)$ converges to the $\ell_{\frac{2}{L}}$ large margin separator defined as

$$\lim_{t \to \infty} \frac{\boldsymbol{\beta}(t)}{|\boldsymbol{\beta}(t)|} = \frac{\boldsymbol{\beta}^*_{diag}}{|\boldsymbol{\beta}^*_{diag}|} \text{ where } \boldsymbol{\beta}^*_{diag} = \arg\min_{\boldsymbol{\beta} \in \mathbb{R}^D} ||\boldsymbol{\beta}||_{\frac{2}{L}} \text{ s.t. } y_n \mathbf{x}_n^\top \boldsymbol{\beta} \geq 1 \quad \forall n.$$

A remarkable consequence of this is that unregularized gradient descent can find sparse classifiers, without any form of explicit regularization. In fact, this inductive bias is even more sparsity-seeking than the typically used $\ell_1$ regularization (see e.g. Koh et al., 2007; Tibshirani, 1996). Figure 2 illustrates this behaviour in a 2D example.

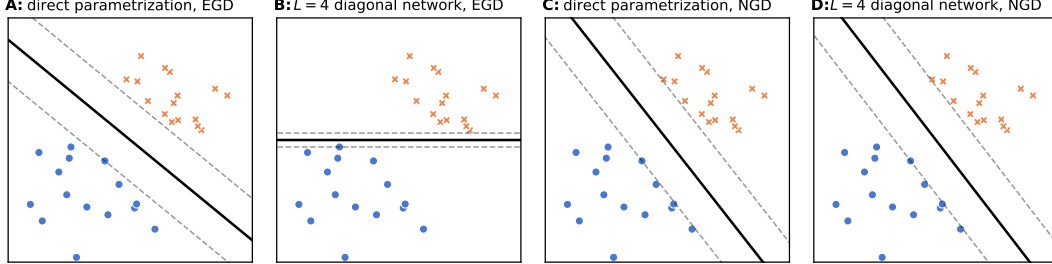

Figure 2: Implicit regularization of EGD and NGD on logistic loss in separable classification. EGD reaches different optima depending on parametrization: fully connected networks reach $\ell_2$ large margin (**A**), while $L$-layer linear diagonal networks reach the $\ell_{\frac{2}{L}}$-large margin solution which favours sparsity (**B**), while $L$-layer linear diagonal networks reach the $\ell_{\frac{2}{L}}$-large m. NGD converges to the same optimum irrespective of the parametrization (**C, D**).

## 2.2 MATRIX COMPLETION VIA DEEP MATRIX FACTORIZATION

The task of matrix completion involves recovering an unknown matrix $\boldsymbol{\beta}^* \in \mathbb{R}^{D \times D}$ from a randomly chosen subset of observed entries[2]. The problem is clearly underdefined: there are infinitely many matrices that match the observed entries. It is common to make additional assumptions about $\boldsymbol{\beta}^*$, most commonly that that it has low rank, under which it becomes identifiable.

One approach to matrix completion under the low-rank assumption is based on explicit regularization (e.g. nuclear norm) which leads to a convex optimization problem. Another common approach is matrix factorization using an underparametrized representation $\boldsymbol{\beta} = UV$ where the sizes of $U \in \mathbb{R}^{D \times R}$ and $V \in \mathbb{R}^{R \times D}$ are restricted to ensure $\boldsymbol{\beta}$'s rank is at most $R$. Learning then proceeds by minimizing the non-convex mean-squared reconstruction error in $U, V$ via gradient descent.

Remarkably, Gunasekar et al. (2017) showed that the gradient-based matrix factorization method tends to converge to low-rank solutions even in the overparametrized setting, i.e. when $\boldsymbol{\beta} = W_1 W_2$ where $W_1$ and $W_2$ are full square matrices, without any explicit regularization. This was later extended by Arora et al. (2019), who studied the deep matrix product parametrization of the form $\boldsymbol{\beta} = W_1 W_2 \cdots W_L$. Arora et al. (2019) ran experiments for different matrix completion tasks varying initialization, depth and number of observations and compared them to minimum nuclear norm solution. When the number of observed entries is large gradient descent in deep matrix factorization models tended to the minimum nuclear norm solution. However, in the interesting case of fewer observed entries, the behaviour was different. Gradient descent pre-

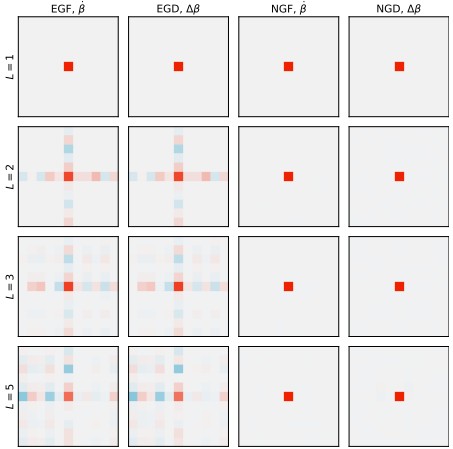

Figure 3: Illustration of the neural tangent kernel in EGF, EGD, NGF and NGD (*left to right*) in matrix factorization models of different depth (*top to bottom*). The algorithms take gradient steps to minimise the squared error on a single observation at the middle of the matrix. Each panel shows how entries of the full $11 \times 11$ matrix move from a random initial state. When $L \geq 2$, Euclidean gradient methods also move entries where there is no observation - enabling implicit regularization towards low-rank solutions. By contrast, and due to invariance, natural gradient methods move only the single entry to match the observation.

ferred solutions with lower effective rank at the expense of higher nuclear norm. From the evolution of the singular values of $\boldsymbol{\beta}$ they also concluded that the implicit regularization is towards low rank that becomes stronger as depth grows.

---

[2]to simplify presentation we assume the matrices are square, but our arguments hold more generally.

## 2.3 NATURAL GRADIENT DESCENT

In the next section we briefly introduce some notation and key properties of natural gradient descent (NGD, Amari, 1997; Pascanu & Bengio, 2013). Intuitively, one can think of NGD as a gradient descent method, but not in the Euclidean space (with the Euclidean metric) of parameters, but instead on the Riemannian manifold of probabilistic models the parameters define (equipped with a different metric). More specifically, let's say that the parameter of interest is $\theta$, where $\theta$ defines a probabilistic model $p(y|\mathbf{x}, \theta)$. We assume that we wish to minimize the log loss under this model, i. e. $l(\theta, \mathbf{x}, y) = -\log p(y|\mathbf{x}, \theta)$ and $\mathcal{L}(\theta) = \sum_{n=1}^{N} l(\theta, \mathbf{x}_n, y_n)$. Then, NGD is usually defined as

$$\theta(t+1) = \theta(t) - \eta F^{-1}(\theta)\nabla_\theta\mathcal{L}(\theta), \text{ where} \tag{1}$$

$$F(\theta) = \mathbb{E}_X[\mathbb{E}_{Y|X;\theta}[\nabla_\theta\mathcal{L}(\theta)\nabla_\theta^\top\mathcal{L}(\theta)]] \tag{2}$$

is the average Fisher information matrix and $\eta$ is the step size. In the above definition, $\mathbb{E}_{Y|X;\theta}$ is taken over the distribution specified by $\theta$, but distribution with respect to which the expectation $\mathbb{E}_X$ is calculated can be arbitrarily chosen. In this article we use the empirical distribution of training data, though other choices are possible (Pascanu & Bengio, 2013). We will also consider natural gradient flow (NGF) the continuous limit of NGD, analogously defined as

$$\dot\theta = -F^{-1}(\theta)\nabla_\theta\mathcal{L}(\theta). \tag{3}$$

We also note, that $F(\theta)$ is not generally invertible, and indeed it will not be in some of the cases we will consider. Therefore, it is more correct to define NGF as any trajectory $\theta_t$ which satisfies

$$F(\theta)\dot\theta = -\nabla_\theta\mathcal{L}(\theta). \tag{4}$$

The natural gradient direction is thus only unique within the eigenspace of $F(\theta)$. Of all natural gradient directions, one common choice is to use the Moore-Penrose pseudoinverse of $F$:

$$\dot\theta = -F^+(\theta)\nabla_\theta\mathcal{L}(\theta). \tag{5}$$

We have seen how in EGD, different parametrization of the same problem leads to drastically different trajectories and optima. However, NGD with infinitesimally small learning rate (i. e. NGF) always follows the same trajectory in model-space and this finds the same optimum, irrespective of how it is parametrized, provided that the parametrization is smooth and locally invertible. Below we formally state this property Amari (1997), alongside a short proof for illustration.

**Statement** (Invariance of NGF under reparametrization). *Let $\mathbf{w}$ and $\theta$ be two parameter vectors related by the mapping $\theta = \mathcal{P}(\mathbf{w})$ and consider natural gradient flow in $\mathbf{w}$. Assume that (1) the Jacobian $J = \frac{\partial\theta_t}{\partial\mathbf{w}_t}$ and (2) $F(\theta_t)$ are both full rank for all $t$. If $\mathbf{w}_t$ follows natural gradient flow starting from $\mathbf{w}_0$ then $\theta_t = \mathcal{P}(\mathbf{w}_t)$ follows NGF, i. e. it solves $\dot\theta_t = -F(\theta_t)^+\nabla_{\theta_t}\mathcal{L}(X, \theta_t)$.*

## 3 NATURAL GRADIENTS UNDER LOGISTIC LOSS ON SEPARABLE DATA

We have seen in Section 2.1 that when trained on separable data with the logistic loss EGD tends to converge to large margin classifiers. To illustrate how NGD differs, we first prove an invariance property which, as we will see, rules out large margin behaviour. We state this property separately when $N < D$ and when $N \geq D$ in the theorems that follow. We denote the number of data points with $N$ and the number of input features with $D$.

**Theorem 1.** *Let's assume, that $N < D$, $X$ is full rank and $A$ is an invertible $D \times D$ matrix. Let $\boldsymbol{\beta}_t = \boldsymbol{\beta}_t(X, \mathbf{y})$ be the trajectory of NGF and $\boldsymbol{\beta}'_t = \boldsymbol{\beta}_t(XA^\top, \mathbf{y})$ (the trajectory of NGF on data $XA^\top$). Then $X\boldsymbol{\beta} = XA^\top\boldsymbol{\beta}'$ (with the assumption that $\boldsymbol{\beta}$ and $\boldsymbol{\beta}'$ have equivalent initial conditions).*

*Proof sketch.* We use the notation $\mathbf{s} = X\boldsymbol{\beta}$ and $\mathbf{s}' = XA^\top\boldsymbol{\beta}$ and prove that $\mathbf{s}_t = \mathbf{s}'_t$. The full proof can be found in Appendix C.1.

**Theorem 2.** *Let $\boldsymbol{\beta}_t(X, \mathbf{y})$ be the trajectory of NGF and let $A$ be a $D \times D$ invertible transformation. If $N \geq D$, $X$ has full rank and we consider NGF on the transformed data $XA^\top$, then $A^\top\boldsymbol{\beta}_t(XA^\top, \mathbf{y}) = \boldsymbol{\beta}_t(X, \mathbf{y})$ (with the assumption that $\boldsymbol{\beta}$ and $\boldsymbol{\beta}'$ have equivalent initial conditions).*

*Remark.* When $N \geq D$ and X is full rank, the size of $F(\boldsymbol{\beta})$ is $D \times D$ and its rank is D, therefore the Fisher information matrix of $\boldsymbol{\beta}$ is invertible.

*Proof sketch.* First let's say $\boldsymbol{\beta}'_t = \boldsymbol{\beta}_t(XA^\top, \mathbf{y})$ and $\boldsymbol{v}^\top = \boldsymbol{\beta}'^\top A$. Then we prove the following:

$$\nabla_{\boldsymbol{\beta}'}\mathcal{L}(y_n\boldsymbol{\beta}'^\top A\mathbf{x}_n) = A\nabla_{\boldsymbol{v}}\mathcal{L}(y_n\boldsymbol{v}^\top\mathbf{x}_n) \quad \text{and} \quad F(\boldsymbol{\beta}') = AF(\boldsymbol{v})A^\top. \tag{6}$$

Hence we get:

$$\dot{\boldsymbol{\beta}}' = F(\boldsymbol{\beta}')^{-1}\nabla_{\boldsymbol{\beta}'}\mathcal{L}(\boldsymbol{\beta}') \quad \text{and} \quad \dot{\boldsymbol{v}} = F(\boldsymbol{v})^{-1}\nabla_{\boldsymbol{v}}\mathcal{L}(\boldsymbol{v}). \tag{7}$$

So if $\boldsymbol{v}$ and $\boldsymbol{\beta}'$ have the same initialization, then $\boldsymbol{v}_t = \boldsymbol{\beta}'_t$. Full proof can be found in Appendix C.2.

**Conclusion.** *Let $u_t(X, \mathbf{y})$ denote the trajectory of $X\boldsymbol{\beta}_t$, which is the linear function $\boldsymbol{\beta}_t^\top\mathbf{x}$ evaluated at each of the datapoints $\mathbf{x}_n$. Then $u_t(XA^\top, \mathbf{y}) = u_t(X, \mathbf{y})$.*

*Proof.* $$u_t(XA^\top, \mathbf{y}) = XA^\top\boldsymbol{\beta}_t(XA^\top, \mathbf{y}) = X\boldsymbol{\beta}_t(X, \mathbf{y}) = u_t(X, \mathbf{y})$$

One special case of this invariance property is invariance to scaling the dimensions of input data (when $A$ is diagonal). Imagine we scale any dimension by a constant $a$, NGF counteracts it by scaling the corresponding coordinate of $\beta$ by $a^{-1}$. We see now why this rules out characterising implicit regularization of NGD as minimizing non-data-dependent norms of $\boldsymbol{\beta}$. In particular, it rules out the $\ell_p$ large-margin behaviour we have seen in EGD.

*Remark.* Let A be a $D \times D$ invertible transformation and let $\boldsymbol{\beta}^*(X, \mathbf{y})$ be the $\ell_2$ large margin solution, *i.e.* $\boldsymbol{\beta}^*(X, \mathbf{y}) = \operatorname{argmin}\|\boldsymbol{\beta}\|_2$ subject to $y_n\boldsymbol{\beta}^\top\mathbf{x}_n \geq 1 \,\forall n$. Then the $\ell_2$ large margin classifier does not have the invariance property, namely there exists a dataset $(X, y)$ and a transformation A such that $A^\top\boldsymbol{\beta}_t^*(XA^\top, \mathbf{y}) \neq \boldsymbol{\beta}_t^*(X, \mathbf{y})$. We include a proof by counterexample in Appendix D.

Having ruled out norm-based implicit regularization, it's natural to consider other statistical methods that exhibit invariance under invertible data transformations. One candidate is ridge-less regression or ordinary least squares (OLS), whose parameter is given by the formula $\boldsymbol{\beta}_{\text{OLS}} = (X^\top X)^{-1}X^\top y$. As it turns out, the connection between NGD in linear regression and the OLS estimate run deeper than sharing this invariance property.

**Theorem 3.** *If $N < D$ and $X$ is full rank, if parameters $\boldsymbol{\beta}_t$ of a linear model follow natural gradient flow under logistic loss, the logits $\mathbf{s}_t = X\boldsymbol{\beta}_t$ follow an asymptotically linear trajectory with direction vector $\mathbf{y}$.*

*Remark.* The Fisher information matrix w.r.t. $\boldsymbol{\beta}$ is $F(\boldsymbol{\beta}) = X^\top D(\boldsymbol{\beta})X$, where $D(\boldsymbol{\beta})$ is diagonal with positive elements on the diagonal. We see, that $\operatorname{rank}(F) = \operatorname{rank}(X) \leq N$, so $F$ is singular, thus several NGF paths are possible. When $\operatorname{rank}(X) = N$, $\boldsymbol{\beta}$ has $D - N$ degrees of freedom and we did describe $\boldsymbol{\beta}$ on $N$ dimensions. That's why we consider $\mathbf{s}$ instead of $\boldsymbol{\beta}$.

This Theorem follows from the more general Theorem 4 which we will state later.

Informally, when we have more parameters than datapoints, NGD discovers a solution that interpolates the training labels $\mathbf{y}$ (encoded as $-1$s and $+1$s) perfectly just like ordinary least squares does in this case. Furthermore, if one uses the Moore-Penrose pseudoinverse to calculate the descent direction, i.e. Eqn. (5), then $\boldsymbol{\beta}_t$ converges in direction to the OLS parameter.

In general cases, OLS interpolation and large-margin (LM) methods find qualitatively different solutions in classification tasks. While the LM solution is typically a linear combination of a small subset of training data (the support vectors), in OLS all datapoints are *support vectors*. As shown in (Hsu et al., 2020), under some conditions this difference disappears in the highly overparametrised regime - when $D > N \log N$. An implication of Theorem 3 is that this phenomenon, known as support vector proliferation, occurs in NGF when $D > N$. Thus there is a regime where NGF and EGF find qualitatively different classifiers, with different generalisation properties (Hsu et al., 2020).

Theorem 3 provided useful in the context of linear models but it turns out it is relatively straightforward to extend this to a result which holds for non-linear overparametrized models as well.

**Theorem 4.** *Let $\mathbf{w} \in \mathbb{R}^P$, $P \geq N$ be the parameters of a classifier with logits $\mathbf{s} = s(X; \mathbf{w}) \in \mathbb{R}^N$. If $\mathbf{w}_t$ follows natural gradient flow on the logistic loss with labels $\mathbf{y}$ and the Jacobian $J_t = \frac{\partial\mathbf{s}_t}{\partial\mathbf{w}_t}$ is of full rank, then $\mathbf{s}_t$ grows asymptotically linearly with direction vector $\mathbf{y}$.*

*Remark.* If our network is linear $J = X$, so Theorem 3 is a special case of Theorem 4 indeed.

*Proof sketch.* The main idea is that, since $J$ is full rank, by parametrization invariance of NGD the trajectory of $\mathbf{s}$ is determined by the trajectory of the corresponding $\boldsymbol{\beta}$.

$$\dot{\mathbf{s}} = -F^{-1}(\mathbf{s})\nabla_{\mathbf{s}}\mathcal{L}(\mathbf{s}) \tag{8}$$

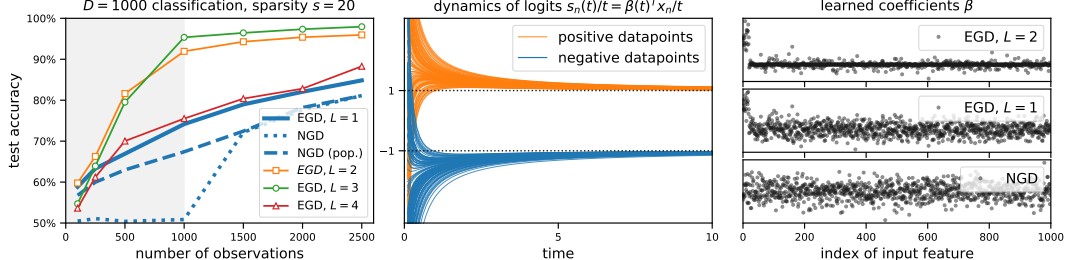

Figure 4: NGD and EGD in a 1000 dimensional sparse classification task, where the ground truth classifier has 20 non-zero components. *Left:* Test accuracy of EGD depends on parametrizaion. When there are there are fewer datapoints than dimensions, EGD with 2 or 3-layer diagonal parametrization can reach up to 90% accuracy. By contrast, when averaging the Fisher infromation on training samples (dotted line) NGD performs at chance level when $N < D$. It performs worse than EGD even when $N \geq D$, or when using the population Fisher calculated on a much larger set of samples (dashed line). *Middle:* Under NGD, when $N < D$, logits of the model grow linearly, proportional to the binary label. *Right:* Coefficient vector $\boldsymbol{\beta}$ learnt by EGD in different architectures and NGD when $N = 2500$: In the 2-layer diagonal network, corresponding to $\ell_1$ implicit regularisation, $\beta$ becomes sparse. In the 1-layer model, the solution is substantially less sparse, but the overall structure is learnt. NGD fails to learn the sparse structure.

Then we can calculate $F(s)$ which turns out to be diagonal, so we have $N$ independent differential equations. We solve them to get the result. The details of the proof can be found in the Appendix C.

## 3.1 EXPERIMENTS

In order to validate and illustrate our findings we have run two main simulations, with results presented in Figures 2 and 4. In both experiments we considered the direct parametrization $\boldsymbol{\beta} = \mathbf{w}$ and the diagonal parametrization $\boldsymbol{\beta} = \mathbf{w}_1 \odot \cdots \odot \mathbf{w}_L$ (Gunasekar et al., 2018) for different depth $L$. In order to run these experiments we needed to implement an efficient algorithm for computing natural gradients in these models: naively calculating and then inverting the Fisher information matrix is computationally inefficient and numerically unstable. We therefore developed an algorithm that exploits the structure in the Fisher information matrix, extending the work of Bernacchia et al. (2018) for diagonal networks. The details of our algorithms can be found in Appendix B.2.

In Experiment 1 we illustrated EGD and NGD in a 2D toy classification dataset. Positive and negative classes were generated such that they are separable by the the axis-aligned separator, but there exists a non-axis-aligned separator with a higher margin. Based on the findings of Gunasekar et al. (2018) we expected EGD to find the large margin solution when $L$ is low, and the axis-aligned solution when $L$ is sufficiently large. The results in panels a and b of Figure 2 confirm these predictions. Figure 2c-d illustrate the parametrization-independence of NGD: it converges to the same solution irrespective of parametrization. The solution is different from both the EGD solutions.

In Experiment 2 we focused on generalization performance. We generated a 1000-dimensional dataset with standard Gaussian $X$, and a sparse ground-truth separator whose first 20 components were set to 1, the rest were 0. Methods with explicit or implicit regularization towards sparse solutions should enjoy good generalization even when $N < D$. Confirming our expectations, we observed that EGD in diagonal parametrizations ($L = 2$, $L = 3$) performed best on this task. The deeper diagonal model ($L = 4$) was on par with the shallow solution, we expect that our 2 million EGD steps were simply not long enough for the implicit regularization to kick in (Moroshko et al., 2020). The NGD solution on the other hand completely fails to generalize when $N < D$ and does relatively poorly even as $N > D$. This catastrophic performance is remedied by averaging the Fisher information on a larger dataset - i. .e. using the population Fisher (Amari et al., 2020), but even this variant of NGD fails to match the performance of EGD. The middle panel of Figure 4 validates the predictions of Theorem 4: logits from the model converge to $t\mathbf{y}$. Finally, the right-hand panels of Figure 4 show that NGD was unable to identify the sparse structure, which the diagonal model infers best, and even the shallow model approximately finds.

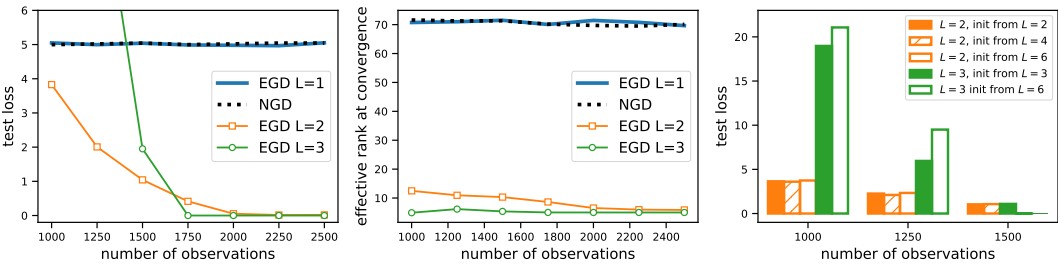

Figure 5: Performance of unregularized EGD and NGD in rank-5 matrix completion tasks using different architectures. *Left and Middle:* Using deep matrix product parametrizations with $L \geq 2$ layers, EGD can reach low training error and identify low-rank solutions even when the number of observations is small. By contrast, NGD in the same problem works similarly to EGD in the naive parametrization and fails to generalize completely. *Right:* 2 (orange) and 3 (green) layer models were initialized by collapsing randomly initialized deeper models to test the effect of initialization separately from the effect of EGD dynamics. Initialization plays a negligible role in the inductive bias of EGD in deep matrix factorization.

## 4 MATRIX COMPLETION WITH NATURAL GRADIENT DESCENT

As we have seen in Section 2.2, EGD in the deep matrix product parametrization $\boldsymbol{\beta} = W_1 \cdots W_L$ converges to low-rank solutions. However, when $L = 1$, i.e. when we run EGD directly on $\boldsymbol{\beta}$, the solution we find is trivial: entries of $\boldsymbol{\beta}$ where we have observation will converge to the observed value, while other entries won't move. Due to parameter-invariance, NGD cannot differentiate between parametrizations of different depth, it is natural to expect that it will fail the same way as EGD does when $L = 1$. Let's look at NGD in matrix completion.

In matrix completion we minimize the squared reconstruction error, which corresponds to the log loss in an isotropic Gaussian observation model with $\boldsymbol{\beta}$ as mean. In a Gaussian model, the Fisher Information Matrix of $\boldsymbol{\beta}$ becomes $F(\boldsymbol{\beta}) = \frac{1}{\sigma_n^2} I$, where $\sigma_n^2$ is the observation noise. The observation noise $\sigma_n^2$ is assumed a constant, and is inconsequential here as it cancels with the $\frac{1}{\sigma_n^2}$ term in the log loss. Consequently, without loss of generality, we can consider $F(\boldsymbol{\beta})$ the identity.

**Statement.** *Let's apply NGF for the problem of matrix completion. EGF in the direct parametrization ($\boldsymbol{\beta} = \mathbf{w}$) is equivalent to NGF under any parametrization $\theta$ for which $J = \frac{\partial \mathbf{w}_t}{\partial \theta_t}$ is full rank.*

The proof of the statement can be found in Appendix E. This implies that NGF will completely fail to generalize, i.e. make an accurate prediction of any unobserved entry of the matrix.

Figure 3 illustrates the key property of the dynamics which allows EGD to generalize in deeper parametrizations. Each panel shows values of the neural tangent kernel (NTK) (Jacot et al., 2018), its equivalent object for NGF called the natural NTK (Rudner et al., 2019), or their discretized versions. For matrix factorization the NTK $K(\theta)$ is a $(D \times D) \times (D \times D)$ tensor which depends on the parameters $\theta$ where $k_{i,j,k,l}(\theta)$ measures how much the entry $\beta_{i,j}$ moves in reaction to a negative loss gradient w.r.t. $\beta_{k,l}$. In these visualizations, we set $D = 11$, and we plot the heatmap of $k_{i,j,5,5}$. We can see that when we parametrize $\boldsymbol{\beta}$ directly, the NTK is simply the identity, only the entry $\beta_{5,5}$ moves. However, when $L = 2$, EGD can now respond to the gradient signal at $\beta_{5,5}$ by moving entries in the fifth row of $W_1$ or in the fifth column of $W_2$. This, in turn, might result in moving $\beta_{i,5}$ or $\beta_{5,i}$ as well. This explains the cross pattern seen in Figure 3 first panel in the second row. This non-identity NTK is what allows generalization to happen as 'information flows' from observations to unobserved entries of $\boldsymbol{\beta}$. However, in NGF, the natural NTK remains the identity irrespective of parametrization. This is true even in the approximately invariant NGD.

For our Matrix Factorization experiments we had to develop a scalable and numerically stable algorithm for computing the natural gradient. We did this by extending the algorithm of Bernacchia et al. (2018) to matrix factorization. Exploiting the structure of the Jacobian in the deep matrix product parametrization ($\boldsymbol{\beta} = W_1 \cdots W_L$) we calculate the natural gradient w.r.t. $W_l$ as $\tilde{\nabla}_{W_l} \mathcal{L} = \frac{1}{L} B_l^\top + \tilde{\nabla}_{\boldsymbol{\beta}} \mathcal{L} A_l^+$, where $A_l = \prod_{i=1}^{l-1} W_i$ and $B_l = \prod_{i=l+1}^{L} W_i$. We note that $A_i$ and

$B_i$ are matrices that are readily computed during the forward and backward pass of reverse-mode automatic differentiation of the loss. The details of the derivation can be found in Appendix B.4.

Using this algorithm, in Figure 5 we experimentally verify that NGD finds a trivial optimum in deep matrix product parametrizations of varying depth. We follow the experimental setup of Arora et al. (2019) and reproduce their results for EGD. We performed an extensive grid search of hyper-parameters and found no setting where NGD would achieve non-trivial performance.

## 5 SUMMARY AND DISCUSSION

Inductive biases of gradient-based learning are driven to a large extent by the way we parametrize our hypothesis. Natural gradient descent (NGD), on the other hand, ignores the parametrization and implicitly optimizes over the manifold of hypothesis. This invited the question whether NGD exhibits any of the useful implicit regularization that EGD has been shown to have. We characterized the behaviour of NGD over logistic loss, and found that in the overparametrized regime, NGD converges to the ordinary least squares interpolant of training labels. This is in contrast with the large-margin-type behaviour EGD exhibits. In experiments we found that in the models we studied, NGD fails to generalize as well as EGD with the right parametrization.

### 5.1 OTHER RELATED WORK

**Approximate NGD algorithms:** Since exact NGD is computationally prohibitive, a great deal of research has been devoted to developing approximate NGD algorithms for deep leaning: K-FAC Grosse & Martens (2016); Martens & Grosse (2015) exploits the approximately Kronecker structure of the Fisher information matrix, while, while Bernacchia et al. (2018) start from exact gradient descent in linear neural networks and then apply the formula verbatim to the non-linear case. Another line of work aims at improving the invariance properties of NGD algorithms bringing them closer to ideal of NGF (Song et al., 2018; Luk & Grosse, 2018). Our motivation differs in that are not focused on designing better NGD algorithms, instead we raise the question whether closer approximation of NGF is desirable in the first place. In order to perform experiments that validate our findings we develop efficient exact natural gradient descent algorithms in overparametrized linear models extending the work of Bernacchia et al. (2018).

**Convergence Rates for NGD:** The main reason for using NGD in deep learning is the intuitive notion it might speed up convergence by virtue of being invariant to parametrization (Amari, 1997; Pascanu & Bengio, 2013; Martens, 2014). This intuition is backed up by theory: Amari (1998) proved fast convergence on a quadratic loss; Bernacchia et al. (2018) proved fast convergence for deep linear models under quadratic loss; more recently, Zhang et al. (2021) gave a proof of fast convergence which holds for a broad class of overparametrized networks and also extends to K-FAC; Rudner et al. (2019) analysed NGD in the neural tangent kernel (NTK) regime. Our work differs in that our primary interest is not whether NGD converges fast, but to better understand and illustrate possible trade-offs between fast convergence and generalization.

**Generalization of NGD:** Wilson et al. (2017) were the first to propose that faster convergence may come at the cost of diminished generalization performance in deep learning. Much like our work, Wilson et al. (2017) provided illustrative examples where different methods reach qualitatively different solutions. They focused on adaptive learning rate algorithms like Adam, but due to the connections between Adam and the empirical Fisher information, one might speculate that their findings would extend to NGD as well Zhang et al. (2019) argued against the notion that NGD may not generalize well, and supported their argument with a generalization bound which holds for both NGD and EGD. However, generalization bounds often fail to predict the empirically observed performance of deep learning (Jiang et al., 2019, see e. g.). In a setting most closely resembling our work Amari et al. (2020) studied generalisation of preconditioned GD for minimising squared loss and found that the optimal preconditioner depends on several factors: EGD generalises better for clean labels, but in scenarios like misspecification or when the labels are noisy, NGD may have an advantage. Finally, Wadia et al. (2021) argued that second order information of the input data - which some second-order optimisation methods can't utilize well, is key to good generalisation in some neural network architectures. This general connection is related to our Theorems 1 and 2.

### 5.2 Q&A

*Q: How about stochastic gradients?* Following Gunasekar et al. (2017; 2018); Arora et al. (2019) we analysed only full-batch gradient descent. This allowed us to prove properties of gradient flow, i. e. in

the limit of infinitesimally small learning rates, which is not a meaningful limit in SGD. This line of work demonstrates that useful inductive biases exist in gradient-based learning even in the absence of gradient noise. Indeed, recent empirical evidence suggests that stochasticity may not be necessary for good generalization in deep networks (see e. g. Geiping et al., 2021). In practice, we expect the question of generalization to be complex, with multiple factors like stochasticity or parametrization-dependence playing a role. We propose that analysing NGD is a useful tool in understanding this complex interplay, as it acts as a form of ablation by eliminating parametrization-dependence.

*Q: Does this mean NGD does not generalize well?* Not necessarily. We show that there are cases where it does not, but it is possible that in other situations the inductive biases of NGD are more helpful than those of EGD + parametrization, especially when trained on large data. Intuitively, our theorems suggest that NGD may be *too efficient* at minimising the training loss at the cost of poorer generalisation. However, in our experiments we saw that averaging the Fisher information matrix over test data may remedy this, which would be in line with the practical recommendation of Pascanu & Bengio (2013). Empirical evidence for generalization in exact NGD is sparse due to the computational cost. Some works report good test performance using approximate methods (Grosse & Martens, 2016; Bernacchia et al., 2018) or small models (Pascanu & Bengio, 2013), but since the focus in these works was on demonstrating the usefulness of new methods, it is questionable how thorough these comparisons were. Zhang et al. (2019); Amari et al. (2020) studied generalisation of natural gradient methods theoretically in limited settings and provided some empirical evidence to support their claims. A systematic empirical investigation similar to (Wilson et al., 2017) may be more informative on this question.

*Q: Does initialization play a role?* Changing the parametrization may influence generalisation in at least three ways: (1) initialization, (2) training dynamics, and (3) constraining the hypothesis space. As weights are often initialized from a parameter-wise independent distribution, these may give rise to a non-trivial and parameter-dependent initial distribution in hypothesis-space. Valle-Pérez et al. (2018) argued that in deep networks, this manifests as a form of simplicity bias. In our models, initialisation has a simlicity bias, too: if matrices $W_1, \ldots, W_L$ are drawn from an isotropic Gaussian, their product $\beta$ will be effectively low-rank with an increasing probability as $L$ increases. By replacing EGD by NGD, we only eliminate the influence of parametrization on training dynamics, but the effects of initialization remain. It is therefore important to disentangle relative importance of initialisation (1), and parameter-dependent dynamics (2). To this end, we designed a set of additional experiments, where we controlled the effect of initialization separately from the effects through dynamics. We initialised deep matrix factorisation models by drawing each component matrix $W_1$ as a product of independent Gaussian matrices, then ran EGD. Thus, we were able to create models behaving like a $L = 6$ layer model at initialization but $L = 2$ layer model during training. We found that the effect of initialization on generalization performance was negligible compared to the effects of training dynamics (Figure 5.c), at least in deep linear models. We further note that initialization plays a very important role in the limit of infinitely wide networks, too, where initialization scale determines whether the network behaves like a linear kernel machine, or more like the behaviour we describe in finite networks here (Woodworth et al., 2020).

*Q: What if you calculate Fisher information on test data?* Pascanu & Bengio (2013) noted that in deep learning, averaging the Fisher information over test data, rather than training data seemingly improves performance. In our theorems and experiments we assume averaging over the training data, sometimes referred to as the sample Fisher information (see e. g. Amari et al., 2020) as this makes our proofs tractable. In our high-dimensional sparse classivication experiment in Figure 4 we tested the performance of NGD when the Fisher information is averaged over a large number of samples, called the population Fisher, and we found that generalisation performance improved, but still did not match that of EGD, especially when sparsity-inducing diagonal parametrisations are used.

*Q: What about other forms of natural gradients?* In addition to the *Fisher-Rao* natural gradients that we consider here, there are other forms of natural gradients, such as those based on the Wasserstein metric (Li & Montufar, 2018; Arbel et al., 2019). When considering this broader family of natural gradient descent, it is natural to ask if the choice of metric may give rise to different inductive biases in NGD similarly to how different parametrizations effect EGD differently. We think this is a fertile area for future research.

REPRODUCIBILITY STATEMENT

Python code to reproduce our results (including all Figures except Figure 1) can be found in the following (anonymized) git repository which contains unit tests and documentation: `https://anonymous.4open.science/r/deeplinear-2F10`

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

## A USEFUL LEMMAS

We will need the following lemma in the proof of Theorem 1,4.

**Lemma 1.** *If we solve a separable classification problem with natrual gradient flow with separator $\boldsymbol{\beta}$ and output $\mathbf{s}$, then the gradient and the Fisher information matrix are the following (in case of linear network this means $\mathbf{s} = X\boldsymbol{\beta}$):*

$$[\nabla_{\mathbf{s}}\mathcal{L}(\mathbf{s})]_i = -y_i(1 - \phi(y_i\mathbf{s}_i)) \tag{9}$$

$$[F(\mathbf{s})]_{i,j} = \delta_{i,j}\phi(\mathbf{s}_i)(1 - \phi(\mathbf{s}_i)) \tag{10}$$

*Proof.* First note that $\phi(u) = \frac{1}{1+e^{-u}}$ and $\phi(-u) = 1 - \phi(u) = \frac{e^{-u}}{1+e^{-u}}$.

$$[\nabla_{\mathbf{s}}\mathcal{L}(\mathbf{s})]_i = \frac{\partial\mathcal{L}}{\partial s_i} = \frac{\partial}{\partial s_i}\sum_{n=1}^N \log(1 + e^{-y_n s_n}) = \frac{-y_i e^{-y_i s_i}}{1 + e^{-y_i s_i}} = -y_i(1 - \phi(y_i s_i)) \tag{11}$$

Using Equation (11) we get the following:

$$[F(\mathbf{s})]_{i,j} = [\mathbb{E}_{\mathbf{y}}[\nabla_{\mathbf{s}}\mathcal{L}(\mathbf{s})\nabla_{\mathbf{s}}^{\top}\mathcal{L}(\mathbf{s})]]_{i,j} = [\mathbb{E}_{\mathbf{y}}[y_i(1 - \phi(y_i s_i))y_j(1 - \phi(y_j s_j))]]_{i,j} =$$

$$= \begin{cases} \mathbb{E}_{\mathbf{y}}[(1 - \phi(y_i s_i))^2] & if \quad i = j \\ \mathbb{E}_{y_i}[y_i(1 - \phi(y_i s_i))]\mathbb{E}_{y_j}[y_j(1 - \phi(y_j s_j))] & if \quad i \neq j \end{cases} =$$

$$= \begin{cases} \phi(s_i)(1 - \phi(s_i)) & if \quad i = j \\ \mathbb{E}_{y_i}[y_i(1 - \phi(y_i))]\mathbb{E}_{y_j}[y_j(1 - \phi(y_j))] & if \quad i \neq j \end{cases} \tag{12}$$

Now we get the following:

$$\mathbb{E}_{y_i}[y_i(1 - \phi(y_i s_i))] = \phi(s_i)(1 - \phi(s_i)) - (1 - \phi(s_i))(1 - \phi(-s_i)) = 0 \tag{13}$$

Hence we get:

$$[F(\mathbf{s})]_{i,j} = \delta_{i,j}\phi(s_i)(1 - \phi(s_i)). \tag{14}$$

$\square$

The next lemma is essential in all computation connected to matrix completion with matrix factorization.

**Lemma 2.** *If we assume that the product matrix $\boldsymbol{\beta}$ comes from a Gaussian distribution with fixed $\sigma_n I$ standard deviation and $\mu$ mean, then the Fisher information matrix of the product matrix in matrix factorization is $F(\boldsymbol{\beta}) = \frac{1}{\sigma_n^2} I$.*

*Proof.* Because of the assumption:

$$p(X|\theta) = \mathcal{N}(X|\mu, \sigma_n I), \tag{15}$$

where $\theta$ is the parameters of the model $(\mu, \sigma_n)$.

$$\nabla_\theta \log p(X|\theta) = \nabla_\theta \log \left( \frac{1}{\sigma_n \sqrt{2\pi}} e^{-\frac{(X-\mu)^2}{2\sigma_n^2}} \right) = \nabla_\mu \left( \log\left( \frac{1}{\sigma_n \sqrt{2\pi}} \right) - \frac{(X-\mu)^2}{2\sigma_n^2} \right) = \frac{(X-\mu)}{\sigma_n^2}, \tag{16}$$

therefore we can compute the Fisher as

$$F(\boldsymbol{\beta}) = \mathbb{E}_{X \sim p(X|\theta)} \left[ \nabla_\theta \log p(X|\theta) \nabla_\theta^\top \log p(X|\theta) \right] = \mathbb{E}_{X \sim p(X|\theta)} \left[ \frac{(X-\mu)(X-\mu)^\top}{\sigma_n^4} \right] = \frac{\sigma_n^2 I}{\sigma_n^4} = \frac{1}{\sigma_n^2} I \tag{17}$$

$\square$

# B   EXACT NATURAL GRADIENTS IN LINEAR MODELS

## B.1   SIMPLE LINEAR MODEL LOGISTIC LOSS

To obtain the natural gradient $\tilde{\nabla}_{\boldsymbol{\beta}} \mathcal{L}$ with respect to $\boldsymbol{\beta}$, we have to solve the following linear system:

$$F(\boldsymbol{\beta}) \tilde{\nabla}_{\boldsymbol{\beta}} \mathcal{L} = \nabla_{\boldsymbol{\beta}} \mathcal{L}, \tag{18}$$

where $F(\boldsymbol{\beta})$ is the Fisher information matrix and $\nabla_{\boldsymbol{\beta}} \mathcal{L}$ is the (Euclidean) gradient. Under the logistic loss the Fisher information matrix becomes

$$F(\boldsymbol{\beta}) = X^\top \operatorname{diag}[\phi(X\boldsymbol{\beta}) \odot \phi(-X\boldsymbol{\beta})]X, \tag{19}$$

where $\phi$ is the logistic sigmoid which is applied elementwise to vector arguments and $\odot$ denotes elementwise product. The gradient of the logistic loss is as follows:

$$\nabla_{\boldsymbol{\beta}} \mathcal{L} = -(y \odot X)^\top \phi(-(y \odot X)\beta) \tag{20}$$

Mathematically, we could use these expressions and solve the linear system Equation (18), however, this would be potentially numerically unstable for reasons outlined below. Let's introduce the notation $\tilde{X} = y \odot X$ and $u = \tilde{X}\boldsymbol{\beta}$ to simplify the formulæ. Due to symmetry, in the Fisher information all occurrences of $X$ can be replaced by $\tilde{X}$. This gives rise to the following expressions for the Fisher information matrix:

$$F(\boldsymbol{\beta}) = \tilde{X}^T \operatorname{diag}[\phi(u) \odot \phi(-u)]\tilde{X} \tag{21}$$

and the gradient:

$$\nabla_{\boldsymbol{\beta}} \mathcal{L} = -\tilde{X}\phi(-u). \tag{22}$$

As the classifier gets better, components of $u$ increase and diverges to $+\infty$. As a consequence both $F(\boldsymbol{\beta})$ and $\nabla_{\boldsymbol{\beta}} \mathcal{L}$ are expected to become small, from the term $\phi(-u)$. This could lead to issues with numerical stability. To solve this, we rewrite both using following identity:

$$\phi(-u) = \frac{1}{1 + e^u} = \frac{e^{-u}}{1 + e^{-u}} = e^{-u}\phi(u) \tag{23}$$

obtaining:

$$F(\boldsymbol{\beta}) = e^{-u_{max}} \tilde{X}^T \operatorname{diag}[e^{-u+u_{max}}\phi^2(u)]\tilde{X} \tag{24}$$

$$\nabla_{\boldsymbol{\beta}} \mathcal{L} = -e^{-u_{max}} \tilde{X} e^{-u+u_{max}}\phi(u), \tag{25}$$

where $u_{max}$ is the largest entry of $u$. We have thus isolated the term responsible for poor numerical performance into a multiplicative term $e^{-u_{max}}$ which we can simply leave out when solving the linear system. The remaining terms are well-behaved even as $u$ increases, provided that the difference between elements of $u$ is not too large.

## B.2 DIAGONAL LINEAR NETWORK UNDER LOGISTIC LOSS

In a diagonal linear network we express $\boldsymbol{\beta} = \mathbf{w}_1 \odot \cdots \odot \mathbf{w}_L$. Here we will discuss how we compute the natural gradient with respect to $\mathbf{w}_1$.

We now solve the following (underdetermined) system of linear equations, which we write using using Einstein summation notation:

$$F(\boldsymbol{\beta})_{i,j} J_{j,l,k} \tilde{\nabla}_{\mathbf{w}_{l,k}} \mathcal{L} = \nabla_{\boldsymbol{\beta}_i} \mathcal{L}, \tag{26}$$

where $J_{j,l,k} = \frac{\partial \boldsymbol{\beta}_j}{\partial \mathbf{w}_{l,k}}$ is the Jacobian of the mapping from $\mathbf{w}$ to $\boldsymbol{\beta}$. In this specific parametrization, most entries of $J$ is non-zero. Let's denote the product of the first $l-1$ weight vectors as $\mathbf{a}_l$ and the product of the last $L - l - 1$ weight vectors as $\mathbf{b}_l$ so we can have:

$$\boldsymbol{\beta}_i = \underbrace{\mathbf{w}_{1,i} \cdots \mathbf{w}_{l-1,i}}_{\mathbf{a}_{l,i}} \mathbf{w}_{l,i} \underbrace{\mathbf{w}_{l+1,i} \cdots \mathbf{w}_{L,i}}_{\mathbf{b}_{l,i}} = \mathbf{a}_{l,i} \mathbf{w}_{l,i} \mathbf{b}_{l,i}. \tag{27}$$

Thus, the Jacobian becomes:

$$J_{i,l,k} = \begin{cases} \mathbf{a}_{l,i} \mathbf{b}_{l,i} & \text{if } i = j \\ 0 & \text{if } i \neq j \end{cases} \tag{28}$$

Substituting this back, we have to solve the following system of equations:

$$F(\boldsymbol{\beta})_{i,j} J_{j,l,k} \tilde{\nabla}_{w_{l,k}} \mathcal{L} = \nabla_{\boldsymbol{\beta}_i} \mathcal{L} \tag{29}$$

$$F(\boldsymbol{\beta})_{i,j} \mathbf{a}_{l,j} \mathbf{b}_{l,j} \tilde{\nabla}_{w_{l,j}} \mathcal{L} = \nabla_{\boldsymbol{\beta}_i} \mathcal{L}. \tag{30}$$

$$\tag{31}$$

To ensure numerical stability, we use the same trick as in B.2.

Since the above system of equations is underdetermined, we could choose different solutions. In our experiments we used the `pytorch.linalg.lstsq` least squares solver which finds the solution with the lowest $\ell_2$ norm.

## B.3 SEPARABLE CLASSIFICATION

First note that in our model ($\forall n \in \{1, 2, \cdots N\}, \phi(s) = \frac{1}{1+e^{-s}}$)

$$p(y_n = 1 | \mathbf{x}_n, \boldsymbol{\beta}) = \frac{1}{1 + e^{-y_n \mathbf{x}_n^\top \boldsymbol{\beta}}} = \phi(-y_n \mathbf{x}_n^\top \boldsymbol{\beta})$$
$$p(y_n = -1 | \mathbf{x}_n, \boldsymbol{\beta}) = 1 - \frac{1}{1 + e^{-y_n \mathbf{x}_n^\top \boldsymbol{\beta}}} = 1 - \phi(-y_n \mathbf{x}_n^\top \boldsymbol{\beta}) \tag{32}$$

So the loss function is ($\forall n \in \{1, 2, \cdots N\}$)

$$\ell(y_n \mathbf{x}_n^\top \boldsymbol{\beta}) = \log(1 + e^{-y_n \mathbf{x}_n^\top \boldsymbol{\beta}}) \tag{33}$$

and

$$\mathcal{L}(\boldsymbol{\beta}) = \sum_{n=1}^{N} \log(1 + e^{-y_n \mathbf{x}_n^\top \boldsymbol{\beta}}) \tag{34}$$

Until now this did not depend on the parametrization. Now look at the parametrizations we used in our article.

If we use a **fully connected network** the gradient is the following:

$$\nabla_{\boldsymbol{\beta}} \mathcal{L}(\boldsymbol{\beta}) = \sum_{n=1}^{N} \nabla_{\boldsymbol{\beta}} \log(1 + e^{-y_n \mathbf{x}_n^\top \boldsymbol{\beta}}) = \sum_{n=1}^{N} \frac{-y_n \mathbf{x}_n e^{-y_n \mathbf{x}_n^\top \boldsymbol{\beta}}}{1 + e^{-y_n \mathbf{x}_n^\top \boldsymbol{\beta}}} =$$
$$= \sum_{n=1}^{N} \frac{-y_n \mathbf{x}_n}{1 + e^{y_n \mathbf{x}_n^\top \boldsymbol{\beta}}} = \sum_{n=1}^{N} -y_n \mathbf{x}_n (1 - \phi(y_n \mathbf{x}_n^\top \boldsymbol{\beta})). \tag{35}$$

The Fisher information matrix is the following:

$$
\begin{aligned}
F(\boldsymbol{\beta}) = \mathbb{E}_X[\mathbb{E}_{Y|X}[\nabla_{\boldsymbol{\beta}}\ell(-y_n\mathbf{x}_n^\top\boldsymbol{\beta})\nabla_{\boldsymbol{\beta}}^\top\ell(-y_n\mathbf{x}_n^\top\boldsymbol{\beta})]] = \\
= \frac{1}{N}\sum_{n=1}^N \mathbb{E}_{Y|X}[\mathbf{x}_n\mathbf{x}_n^\top(1-\phi(y_n\mathbf{x}_n^\top\boldsymbol{\beta}))^2] = \\
= \frac{1}{N}\sum_{n=1}^N \mathbf{x}_n\mathbf{x}_n^\top(\phi(\mathbf{x}_n^\top\boldsymbol{\beta})(1-\phi(\mathbf{x}_n^\top\boldsymbol{\beta}))^2 + (1-\phi(x_n^\top\boldsymbol{\beta}))(1-\phi(-\mathbf{x}_n^\top\boldsymbol{\beta}))^2) = \\
= \frac{1}{N}\sum_{n=1}^N \mathbf{x}_n\mathbf{x}_n^\top(\phi(\mathbf{x}_n^\top\boldsymbol{\beta})(1-\phi(\mathbf{x}_n^\top\boldsymbol{\beta}))^2 + (1-\phi(x_n^\top\boldsymbol{\beta}))\phi^2(\mathbf{x}_n^\top\boldsymbol{\beta})) = \\
= \frac{1}{N}\sum_{n=1}^N \mathbf{x}_n\mathbf{x}_n^\top\phi(\mathbf{x}_n^\top\boldsymbol{\beta})(1-\phi(\mathbf{x}_n^\top\boldsymbol{\beta}))
\end{aligned}
\tag{36}
$$

If we use a **diagonal network** $\boldsymbol{\beta} = \mathbf{w}_1 \odot \mathbf{w}_2 \odot \cdots \odot \mathbf{w}_{L-1} \odot \mathbf{w}_L$, where $\mathbf{w} = \begin{pmatrix} \mathbf{w}_1^\top & \mathbf{w}_2^\top & \cdots & \mathbf{w}_L^\top \end{pmatrix}^\top$. The gradient is the following:

$$
\nabla_{\boldsymbol{\beta}}\mathcal{L} = J^\top\nabla_{\mathbf{w}}\mathcal{L}
\tag{37}
$$

where $J$ (the Jacobian) is the following

$$
J = \begin{pmatrix} \frac{\partial\boldsymbol{\beta}}{\partial\mathbf{w}_1} & \cdots & \frac{\partial\boldsymbol{\beta}}{\partial\mathbf{w}_L} \end{pmatrix}.
\tag{38}
$$

where

$$
\left[\frac{\partial\boldsymbol{\beta}}{\partial\mathbf{w}_n}\right]_{i,j} = \frac{\partial\boldsymbol{\beta}_i}{\partial[\mathbf{w}_n]_j} = \delta_{i,j}\prod_{k=1,k\neq i}^N [\mathbf{w}_k]_i
\tag{39}
$$

The Fisher information matrix is the following:

$$
F(\mathbf{w}) = J^\top F(\boldsymbol{\beta})J
\tag{40}
$$

## B.4 MATRIX FACTORIZATION

Before we compute the natural gradient of matrix factorization let us introduce some notations: $\boldsymbol{\beta} = W_1W_2\ldots W_L$, as before and

$$
\theta = vec(\boldsymbol{\beta}),
\tag{41}
$$
$$
\mathbf{w} = vec(W_1, W_2, \ldots, W_L),
\tag{42}
$$

where *vec* vectorizes the matrices to obtain a column vector. $\theta$ is a reparametrization of $\mathbf{w}$, so $\theta = \mathcal{P}(\mathbf{w})$ and let $J = \frac{\partial\theta}{\partial\mathbf{w}}$. With this notation, let's compute the natural gradient with respect to the parametrization $\mathbf{w}$.

$$
\tilde{\nabla}_{\mathbf{w}}\mathcal{L} = F(\mathbf{w})^{-1}\nabla_{\mathbf{w}}\mathcal{L} = (J^\top F(\theta))J)^{-1}(J^\top\nabla_\theta\mathcal{L}) = J^{-1}F(\theta)^{-1}\nabla_\theta\mathcal{L}
\tag{43}
$$

We use the assumption that $J$ is full rank and because of $F(\theta) = I$ is invertible $(J^\top F(\theta))J)^{-1} = J^{-1}F(\theta)^{-1}J^{-\top}$. Thus, the natural gradient simplifies to

$$
\tilde{\nabla}_{\mathbf{w}}\mathcal{L} = J^{-1}\nabla_\theta\mathcal{L}
\tag{44}
$$

and multiplying by $J$ we obtain

$$
J\tilde{\nabla}_{\mathbf{w}}\mathcal{L} = \nabla_\theta\mathcal{L}.
\tag{45}
$$

We can consider the Jacobian like L consecutive matrices

$$
J = [J_1 J_2 \ldots J_L]
\tag{46}
$$

where $J_i = \frac{\partial\theta}{\partial vec(W_i)}$, and note that $\nabla_\theta\mathcal{L} = vec(\nabla_{\boldsymbol{\beta}}\mathcal{L})$ and $\tilde{\nabla}_{\mathbf{w}}\mathcal{L} = vec(\tilde{\nabla}_{W_1,W_2,\ldots W_L}\mathcal{L})$. Rewrite equation 45:

$$
Jvec(\tilde{\nabla}_{W_1,W_2,\ldots W_L}\mathcal{L}) = vec(\nabla_{\boldsymbol{\beta}}\mathcal{L}).
\tag{47}
$$

If we solve the following equation for $i = 1, \ldots L$, then the concatenation of vectors $vec(\tilde{\nabla}_{W_i}\mathcal{L})$ will solve equation 47 as well.

$$J_i vec(\tilde{\nabla}_{W_i}\mathcal{L}) = \frac{1}{L}vec(\nabla_{\boldsymbol{\beta}}\mathcal{L}) \tag{48}$$

Let $A_i = W_1 W_2 \ldots W_{i-1}$ and $B_i = W_{i+1}W_{i+2}\ldots W_L$ and using $\otimes$ notation for the Kronecker product and utilize the property $vec(ABC) = (C^\top \otimes A)vec(X)$ we get

$$J_i = \frac{\partial vec(\boldsymbol{\beta})}{\partial vec(W_i)} = \frac{\partial vec(A_i W_i B_i)}{\partial vec(W_i)} = \frac{\partial (B_i^\top \otimes A_i)vec(W_i)}{\partial vec(W_i)} = B_i^\top \otimes A_i, \tag{49}$$

thus we need to solve

$$(B_i^\top \otimes A_i)vec(\tilde{\nabla}_{W_i}\mathcal{L}) = \frac{1}{L}vec(\nabla_{\boldsymbol{\beta}}\mathcal{L}) \tag{50}$$

for $vec(\tilde{\nabla}_{W_i}\mathcal{L})$. One can do this by exploiting properties of the Kronecker product and using Moore-Penrose pseudo-inverses as follows:

$$vec(\tilde{\nabla}_{W_i}\mathcal{L}) = \frac{1}{L}(B_i^{\top +} \otimes A_i^+)vec(\nabla_{\boldsymbol{\beta}}\mathcal{L}) = \frac{1}{L}(B_i^{\top +}\nabla_{\boldsymbol{\beta}}\mathcal{L}A_i^+) \tag{51}$$

We note that when $A_i$ and $B_i$ are near full-rank, using the pseudoinverses may not be numerically stable. Fausett & Fulton (1994) instead proposed a solution based on QR decomposition, and even discussed an approach which extends to the rank deficient case. In practice we found that this was not necessary for ours experiments. As a result, in our implementation we use the formula $\frac{1}{L}B_i^{\top +}\nabla_{\boldsymbol{\beta}}\mathcal{L}A_i^+$ to update the factor matrices with the natural gradient.

## C  PROOF OF THEOREMS

### C.1  PROOF OF THEOREM 1

**Statement.** *Let's assume, that $N < D$, $X$ is full rank and $A$ is an invertible $D \times D$ matrix. Let $\boldsymbol{\beta}_t = \boldsymbol{\beta}_t(X, \mathbf{y})$ be the trajectory of NGF and $\boldsymbol{\beta}'_t = \boldsymbol{\beta}_t(XA^\top, \mathbf{y})$ (the trajectory of NGF on data $XA^\top$). Then $X\boldsymbol{\beta} = XA^T\boldsymbol{\beta}'$.*

*Proof.* Let $\mathbf{s} = X\boldsymbol{\beta}$ and $\mathbf{s}' = XA^T\boldsymbol{\beta}'$. The gradient and the Fisher information matrix are the following (the calculation can be found in Lemma 1).

$$\begin{aligned} [\nabla_{\mathbf{s}}\mathcal{L}(\mathbf{s})]_i &= -y_i(1 - \phi(y_i\mathbf{s}_i)) \\ [F(\mathbf{s})]_{i,j} &= \delta_{i,j}\phi(\mathbf{s}_i)(1 - \phi(\mathbf{s}_i)) \end{aligned} \tag{52}$$

The exact same can be said about $s'$, so $s$ and $s'$ are the solutions of the same differential equations, so if we use the same initialization $s_t = s'_t$. $\qquad\square$

### C.2  PROOF OF THEOREM 2

**Statement.** *Let $\boldsymbol{\beta}_t(X, \mathbf{y})$ be the trajectory of NGF and let $A$ be a $D \times D$ invertible transformation. If $N \geq D$, $X$ has full rank and we consider NGF on the transformed data $XA^\top$, then $A^\top \boldsymbol{\beta}_t(XA^\top, \mathbf{y}) = \boldsymbol{\beta}_t(X, \mathbf{y})$.*

*Proof.* Let $\boldsymbol{\beta}'$ be the trajectory of NGF on the transformed data:

$$\boldsymbol{\beta}'_t = \boldsymbol{\beta}_t(XA^\top, \mathbf{y}). \tag{53}$$

To run NGF on $\boldsymbol{\beta}'$ we need its Fisher information matrix. Note that the Fisher information matrix of linear models with logistic-loss is

$$F(\boldsymbol{\beta}) = X^\top \text{diag}[\phi(X\boldsymbol{\beta}) \odot \phi(-X\boldsymbol{\beta})]X. \tag{54}$$

by Appendix B.1. Note that in this case the rank of the Fisher information matrix is $D$, so it is invertible. Same is true for $\boldsymbol{\beta}'$. Let's compute the Fisher information matrix of $\boldsymbol{\beta}'$.

$$F(\boldsymbol{\beta}') = \frac{1}{N}\sum_{n=1}^{N} \mathbb{E}_{y_n|A\mathbf{x}_n}[\nabla_{\boldsymbol{\beta}'}\ell(y_n\boldsymbol{\beta}'^\top A\mathbf{x}_n)\nabla_{\boldsymbol{\beta}'}^\top \ell(y_n\boldsymbol{\beta}'^\top A\mathbf{x}_n)] \tag{55}$$

First, specify $\nabla_{\boldsymbol{\beta}'}\ell(y_n\boldsymbol{\beta}'^\top A\mathbf{x}_n)$ and use the notation $\boldsymbol{v}^\top = \boldsymbol{\beta}'^\top A$.

$$\nabla_{\boldsymbol{\beta}'}\ell(y_n\boldsymbol{\beta}'^\top A\mathbf{x}_n) = J^\top \nabla_{\boldsymbol{v}^\top}\ell(y_n\boldsymbol{v}^\top\mathbf{x}_n) \tag{56}$$

where $J = \frac{\partial v^\top}{\partial \boldsymbol{\beta}'}$.

$$J_{i,j} = \frac{\partial \boldsymbol{v}_i}{\partial \boldsymbol{\beta}'_j} = \frac{\partial \sum_{k=1}^d \boldsymbol{\beta}'_k A_{k,i}}{\partial \boldsymbol{\beta}'_j} = A_{j,i} \tag{57}$$

Therefore $J = A^\top \Leftrightarrow J^\top = A$ and

$$\nabla_{\boldsymbol{\beta}'}\ell(y_n\boldsymbol{\beta}'^\top A\mathbf{x}_n) = A\nabla_{\boldsymbol{v}}\ell(y_n\boldsymbol{v}^\top\mathbf{x}_n). \tag{58}$$

We now can continue the computation of the Fisher:

$$F(\boldsymbol{\beta}') = \frac{1}{N}\sum_{n=1}^N \mathbb{E}_{y_n|A\mathbf{x}_n}[A\nabla_{\boldsymbol{v}^\top}\ell(y_n\boldsymbol{v}^\top\mathbf{x}_n)\nabla_{\boldsymbol{v}^\top}^\top\ell(y_n\boldsymbol{v}^\top\mathbf{x}_n)A^\top] = $$

$$= A(\frac{1}{N}\sum_{n=1}^N \mathbb{E}_{y_n|\mathbf{x}_n}[\nabla_{\boldsymbol{v}^\top}\ell(y_n\boldsymbol{v}^\top\mathbf{x}_n)\nabla_{\boldsymbol{v}^\top}^\top\ell(y_n\boldsymbol{v}^\top\mathbf{x}_n)])A^\top = AF(\boldsymbol{v})A^\top. \tag{59}$$

Note, that the Fisher of $\boldsymbol{v}$ must be invertible as well from the previous Equation. Let's see the NGF on $\boldsymbol{\beta}'$:

$$\dot{\boldsymbol{\beta}}' = -F(\boldsymbol{\beta}')^{-1}\nabla_{\boldsymbol{\beta}'}\mathcal{L}(\boldsymbol{\beta}'^\top X A^T, y) = -(AF(\boldsymbol{v})A^\top)^{-1}\sum_{n=1}^N \nabla_{\boldsymbol{\beta}'}\ell(y_n\boldsymbol{\beta}'^\top Ax_n) = $$

$$= -(A^\top)^{-1}F(\boldsymbol{v})^{-1}A^{-1}A\sum_{n=1}^N \nabla_{\boldsymbol{v}}\ell(y_n\boldsymbol{v}^\top x_n) = -(A^\top)^{-1}F(\boldsymbol{v})^{-1}\nabla_{\boldsymbol{v}}\mathcal{L}(\boldsymbol{v}) \tag{60}$$

We also have the following (by the Chain Rule):

$$\dot{\boldsymbol{v}} = J\dot{\boldsymbol{\beta}}' = A^\top\dot{\boldsymbol{\beta}}' \tag{61}$$

Now from Equation (60) and (61) we get:

$$\dot{\boldsymbol{v}} = F(\boldsymbol{v})^{-1}\nabla_{\boldsymbol{v}}\mathcal{L}(\boldsymbol{v}) \tag{62}$$

This is the same differential equation as the one $\boldsymbol{\beta}$ is a solution of. So if they are initialized the same way $\boldsymbol{v}_t = \boldsymbol{\beta}_t(X, \mathbf{y})$, so $\boldsymbol{\beta}_t = \boldsymbol{v}_t = A^\top\boldsymbol{\beta}'$. $\square$

## C.3 PROOF OF THEOREM 4

**Statement.** *If $N < D$ and $\boldsymbol{\beta}$ is the separator. Let $\mathbf{s}$ be the output. Let the Jacobi matrix from $\boldsymbol{\beta}$ to $\mathbf{s}$ be $J = \frac{\partial s}{\partial \boldsymbol{\beta}}$. If $J$ is full rank $\mathbf{s}$ is asymptotically linear with direction vector $\mathbf{y}$.*

*Proof.* First let's note that by the invariance property of NGF the trajectory of $\mathbf{s}$ is defined by the trajectory of $\boldsymbol{\beta}$.

$$\dot{\mathbf{s}} = -F^{-1}(\mathbf{s})\nabla_{\mathbf{s}}\mathcal{L}(\mathbf{s}) \tag{63}$$

Let's assume $\mathbf{s}$ is 1-dimensional. In this case $\mathbf{s} = s$, $\mathbf{x}_1 = \mathbf{x}$ and $\mathbf{y} = y$ can be used since we have only one data point. To solve equation (63) we need the gradient and the Fisher information matrix which are the following (the calculation can be found in the Appendix B.2)

$$\nabla_s\mathcal{L}(s) = -y(1 - \phi(ys)) \tag{64}$$

The Fisher information matrix:

$$F(s) = \phi(s)(1 - \phi(s)) \tag{65}$$

Then Equation (63) can be written as:

$$\dot{s} = \frac{y(1 - \phi(ys))}{\phi(s)(1 - \phi(s))} \tag{66}$$

Now we rescale our data points s.t. $\tilde{x} = -x$, so $\tilde{s} = -s$ and $\tilde{y} = -y = 1$. Hence we get:

$$\frac{\partial \tilde{s}}{\partial t} = \frac{1}{\phi(\tilde{s})} \tag{67}$$

Which can be solved and the solution is

$$\log(1 + e^{\tilde{s}}) = t + c \quad \Longleftrightarrow \quad \tilde{s} = \log(e^{t+c} - 1) \tag{68}$$

By equation (68) we get the asymptotic behaviour

$$\lim_{t \to \infty} \frac{\tilde{s}}{t + c} = \lim_{t \to \infty} \frac{\log(e^{t+c} - 1)}{t + c} = \tag{69}$$

$$= \lim_{t \to \infty} \frac{e^{t+c}}{e^{t+c} - 1} = \lim_{t \to \infty} \frac{1}{1 - e^{-(t+c)}} = 1 \tag{70}$$

(From (69) to (70) we use L'Hopital Rule).

Hence we proved Theorem 4. for $N = 1$. Now let's assume, that $N > 1$. Now we write down the gradient again:

$$[\nabla_{\mathbf{s}} \mathcal{L}(\mathbf{s})]_i = -y_i(1 - \phi(y_i s_i)) \tag{71}$$

And the Fisher information matrix:

$$[F(\mathbf{s})]_{i,j} = \delta_{i,j} \phi(s_i)(1 - \phi(s_i)) \tag{72}$$

So now if we substitute in Equation (71) and Equation (72) to Equation (63). We can rescale, so $\tilde{y}_i = 1 \quad \forall i$ as we did in the previous case. Hence we get the following:

$$\frac{\partial \tilde{\mathbf{s}}}{\partial t} = - \begin{pmatrix} \frac{1}{\phi(\tilde{s}_1)(1-\phi(\tilde{s}_1))} & 0 & \cdots & 0 \\ 0 & \frac{1}{\phi(\tilde{s}_2)(1-\phi(\tilde{s}_2))} & \cdots & 0 \\ \vdots & \vdots & \ddots & \vdots \\ 0 & 0 & \cdots & \frac{1}{\phi(\tilde{s}_N)(1-\phi(\tilde{s}_N))} \end{pmatrix} \begin{pmatrix} -(1 - \phi(\tilde{s}_1)) \\ -(1 - \phi(\tilde{s}_2)) \\ \vdots \\ -(1 - \phi(\tilde{s}_N)) \end{pmatrix} = \begin{pmatrix} \frac{1}{\phi(\tilde{s}_1)} \\ \frac{1}{\phi(\tilde{s}_2)} \\ \vdots \\ \frac{1}{\phi(\tilde{s}_N)} \end{pmatrix} \tag{73}$$

Hence we got $N$ independent differential equations which are exactly the same as in the $N = 1$ case. So in each dimension $\tilde{s}$ is asymptotically $t + c$ for some $c$. Hence $\tilde{\mathbf{s}} \approx t\mathbb{1} + \mathbf{c}$, where $\mathbf{c} \in \mathbb{R}^D$ is a constant. So $\mathbf{s} \approx t\mathbf{y} + \mathbf{c}_s$, where $\mathbf{c}_s \in \mathbb{R}^D$ is a constant. $\qquad\square$

## D    COUNTEREXAMPLE FOR THE INVARIANCE OF $\ell_2$ LARGE MARGIN SOLUTION

The counterexample is the following:

$$A = \begin{pmatrix} 1 & 2 \\ -1 & 0 \end{pmatrix}, y = 1 \text{ and } X = (2 \quad -3)$$

Then $\boldsymbol{\beta}^*(X, y) = argmin\|\boldsymbol{\beta}\|_2$ subject to $2\boldsymbol{\beta}_1 - 3\boldsymbol{\beta}_2 \geq 1$, therefore $\boldsymbol{\beta}^*(X, y) = \begin{pmatrix} 0 \\ -\frac{1}{3} \end{pmatrix}$. Furthermore $\boldsymbol{\beta}^*(XA^\top, y) = argmin\|\boldsymbol{\beta}\|_2$ subject to $-4\boldsymbol{\beta}_1 - 2\boldsymbol{\beta}_2 \geq 1$, therefore $\boldsymbol{\beta}^*(XA^\top, y) = \begin{pmatrix} -\frac{1}{4} \\ 0 \end{pmatrix}$, but $A^\top \boldsymbol{\beta}^*(XA^\top, y) = \begin{pmatrix} -\frac{1}{4} \\ -\frac{1}{2} \end{pmatrix} \neq \begin{pmatrix} 0 \\ -\frac{1}{3} \end{pmatrix} = \boldsymbol{\beta}^*(X, y)$.

## E    PROOF OF THE STATEMENT ABOUT THE PARAMETRIZATION INVARIANCE OF NGF

**Statement.** *Let $\mathbf{w}$ and $\theta$ be two parameter vectors related by the mapping $\theta = \mathcal{P}(\mathbf{w})$ and consider natural gradient flow in $\mathbf{w}$. Assume that (1) the Jacobian $J = \frac{\partial \theta_t}{\partial \mathbf{w}_t}$ and (2) $F(\theta_t)$ are both full rank for all $t$. If $\mathbf{w}_t$ follows natural gradient flow starting from $\mathbf{w}_0$ then $\theta_t = \mathcal{P}(\mathbf{w}_t)$ follows NGF, i.e. it solves $\dot{\theta}_t = -F(\theta_t)^+ \nabla_{\theta_t} \mathcal{L}(X, \theta_t)$.*

*Proof.* We use that $F(\mathbf{w}) = J^\top F(\theta)J$ which follows from the definition of $F$:

$$F(\mathbf{w}) = \mathbb{E}_X[\nabla_{\mathbf{w}}\mathcal{L}(X,\mathbf{w})\nabla_{\mathbf{w}}^\top\mathcal{L}(X,\mathbf{w})] = \mathbb{E}_X[J^\top\nabla_\theta\mathcal{L}(X,\theta)\nabla_\theta^\top\mathcal{L}(X,\theta)J] = J^\top F(\theta)J$$

The invariance statement follows:

$$\dot{\theta} = (\mathcal{P}(\dot{\mathbf{w}_\mathbf{t}})) = J\dot{\mathbf{w}}_\mathbf{t} = -JF(\mathbf{w}_t)^+\nabla_{\mathbf{w}_t}\mathcal{L}(X,\mathbf{w}_t) =$$
$$= -JJ^+F(\theta_t)^+(J^\top)^+J^\top\nabla_{\theta_t}\mathcal{L}(X,\theta_t) = -F(\theta_t)^+\nabla_{\theta_t}\mathcal{L}(X,\theta_t).$$

## F  PROOF OF THE STATEMENT ABOUT NGD IN MATRIX COMPLETION

**Statement.** *Let's apply NGF for the problem of matrix completion. EGF in the direct parametrization ($\boldsymbol{\beta} = \mathbf{w}$) is equivalent to NGF under any parametrization $\theta$ for which $J = \frac{\partial\mathbf{w}_t}{\partial\theta_t}$ is full rank.*

*Proof.* First let's consider a parametrization $\theta$ s.t. the direct parametrization $\boldsymbol{\beta} = \mathbf{w}$ ($= \mathcal{P}(\theta)$) and $J = \frac{\partial\mathbf{w}}{\partial\theta}$ is full rank. Then by the invariance property if $\theta_t$ is the solution of the NGF with the arbitrary parametrization, then $\mathbf{w}_t = \mathcal{P}(\theta_t)$ is the solution of:

$$\dot{\mathbf{w}} = -\nabla_{\mathbf{w}}\mathcal{L}(\mathbf{w})$$

Which agrees with the EGF with direct parametrization. □

## G  INVARIANCE PROPERTY OF OLS

We show the same transformation invariance property for OLS that we showed in Theorem 1,2 for NGF. Again, we split the problem into two cases: $N < D$ and $N \geq D$. Note that for the problem $X\boldsymbol{\beta} = y$ the Ordinary least squares solution is $\boldsymbol{\beta} = (X^\top X)^{-1}X^\top y$ if the columns of X is linearly independent.

**Statement.** *Let $N < D$, A is an invertible $D \times D$ matrix. If $\boldsymbol{\beta}$ is the solution of the Ordenary least squares problem for the matrix X and $\boldsymbol{\beta}'$ for $XA^\top$, then $X\boldsymbol{\beta} = XA^\top\boldsymbol{\beta}'$.*

*Proof.* Immediately follows from the definition of the problems: $X\boldsymbol{\beta} = y$ and $XA^\top\boldsymbol{\beta}' = y$. □

**Statement.** *Let $N \geq D$, X has full rank and A is an invertible $D \times D$ matrix. If $\boldsymbol{\beta}$ is the solution of the OLS problem for the matrix X and $\boldsymbol{\beta}'$ for $XA^\top$, then $A^\top\boldsymbol{\beta}' = \boldsymbol{\beta}$.*

*Proof.*

$$A^\top\boldsymbol{\beta}' = A^\top((XA^\top)^\top XA^\top)^{-1}(XA^\top)^\top y = A^\top A^{\top-1}(X^\top X)^{-1}A^{-1}AX^\top y = \boldsymbol{\beta}$$

□

## H  SUPPLEMENTARY FIGURES

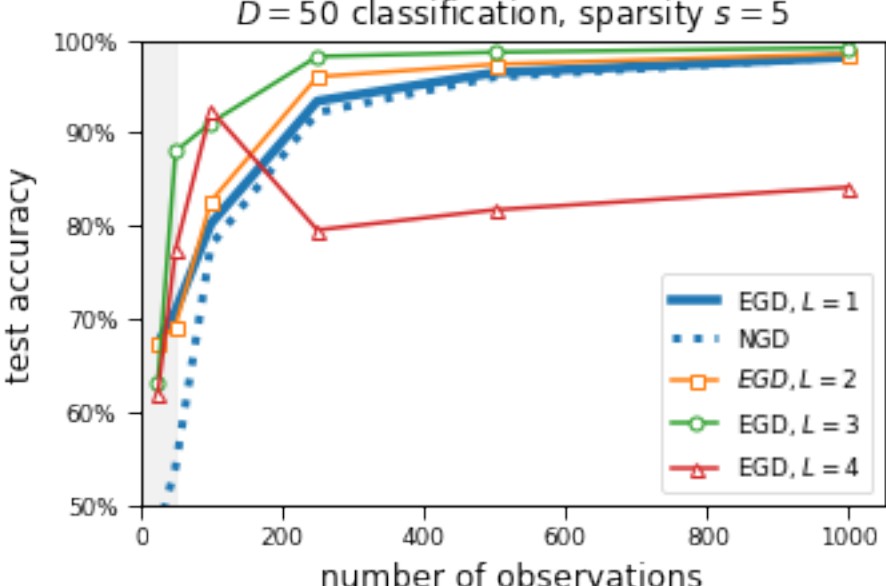

Figure 6: During peer review, reviewers requested a lower dimensional variant of the experiment reported in Figure 4. Instead of 1000 dimensions, in this experiment we used $D = 50$, and instead of $S = 20$ non-zero components, the real $\beta$ had $S = 5$ non-zero entries. The experimental setup and hyperparameters were otherwise not changed from Figure 4. The 5-layer diagonal network performs poorly, which is likely a result of sensitivity to hyperparameters, we expect that with additional fine-tuning of the hyperparameters for this experiment, $L = 4$ would do at least as well as the shallow $L = 1$ model.

