# OpenReview forum: "Depth Without the Magic: Inductive Bias of Natural Gradient Descent"
_ICLR.cc/2022/Conference — ICLR 2022 Submitted_

### Official Review · Reviewer_74k1 · 2021-10-31

**Correctness:** 3
**Technical Novelty And Significance:** 3
**Empirical Novelty And Significance:** Not applicable
**Recommendation:** 5
**Confidence:** 3

**Main Review:**

## Strengths
This paper in general is well-written and easy to read.  This paper is a findings paper.

## Weaknesses
There are some issues that could weaken the arguments made in this paper.

### 1. Invariance of NGD
The statement in Sec 2.3 assumes the Fisher information matrix (FIM) is non-singular (full-rank).
The authors should clarify whether the FIMs considered in the Experiments in Sec 3.1 and Sec 4 are indeed non-singular.
If the pseudoinverse is indeed employed, the authors should also show that the invariance of NGD or NGF also holds in singular cases.

### 2. Theorem 4
In Theorem 4, the authors assume that the Jacobian matrix is full-rank.
The authors should give an example of a 2-layer (linear) NN to illustrate how this assumption is satisfied. I also wonder why the FIM is non-singular in this case since Eq 9 implies that the FIM is invertible.


### 3. Meaning of D
Does D mean the number of input features? If so, I wonder whether the authors consider the most common cases when the number of input features (D) << the number of (training) data points (N) << the number of NN parameters.
This question is closely related to pointer 4.
I think all experiments considered should report these three numbers.

### 4. Empirical FIM approximations
In Eq 2, the exact FIM is computed under the expectation w.r.t. x.  In this work, the FIM is computed over a set of training data, which is known as an empirical approximation in statistics. If the number of data points << the number of input features, the empirical approximation of the FIM could be bad (see [1]).
In experiment 2, the authors show that NGD could perform poorly. The authors should report the number of parameters used in this experiment.
According to the caption of Figure 4, the number of training data points is 2500 and the number of input features is 1000.  I do not think the empirical approximation of the FIM over 2500 training data points is good enough since the FIM should be at least a 1000-by-1000 matrix.
I would like to see the performance of NGD when the number of input features is changed from 1000 to 50. For the sparse classification task, the authors may set the first 5 components to be 1 instead of the first 20 components.


### 5. Practical NGD with damping (ridge regression)
In practice, I do not think the pseudoinverse is used due to the high computational cost.
It will be great if the authors can comment on cases when damping is used. Does damping introduce extra inductive bias of NGD?

### 6. Initialization does play a role
NGD is a discretization of NGF. Solving NGF is an initial value problem (IVP) in this setting.
Thus, initialization should play a role such as pre-training. The statement about initialization should be more carefully stated.



 ## References
 [1] Kunstner, Frederik, Philipp Hennig, and Lukas Balles. "Limitations of the empirical Fisher approximation for natural gradient descent." Advances in Neural Information Processing Systems 32 (2019): 4156-4167.




**Summary Of The Paper:**

In this work, the authors study the inductive bias of natural gradient descent in some simple and ideal settings.
The authors argue that the invariance of NGD could lead to a worse generation error compared to standard gradient descent (EGD) in some settings.


**Summary Of The Review:**

This paper in general is well-written and easy to read.  Some arguments made in this work seem to be reasonable. However, there are issues that weaken the arguments. Please address the issues in the main review. I am happy to give a higher rating.

---

> ### Author Response · Authors · 2021-11-18
> **Response to Reviewer 74k1**
>
> Thank you for your detailed and encouraging review. Our responses to your concerns and comments:
>
> >The statement in Sec 2.3 assumes the Fisher information matrix (FIM) is non-singular (full-rank). The authors should clarify whether the FIMs considered in the Experiments in Sec 3.1 and Sec 4 are indeed non-singular. If the pseudoinverse is indeed employed, the authors should also show that the invariance of NGD or NGF also holds in singular cases.
>
> Indeed, in situations where N < D < P, the Fisher information matrix becomes singular in both the D-dimensional and P-dimensional parametrization. However, it is still true that the NGF trajectories will be equivalent within the subspace spanned by training samples. To see this, consider running NGF directly on the N-dimensional logits of the model. The Fisher information matrix here is always non-singular (diagonal). So we can apply our theorem to show that NGF in (D-dimensional) beta space and NGF in the (P-dimensional) w space are both equivalent to NGF in the logits, within the N-dimensional space spanned by data. It is true that technically, when N < P, NGF in the P-dimensional space is underdetermined, and the precise behaviour of NGF on test data may vary depending on the specific solution we choose. Using the pseudoinverse would result in the smallest norm updates in the specific parametrization, so performing NGF with singular Fisher and pseudoinverse could, technically, retain some parametrization-dependent inductive biases.
>
> >In Theorem 4, the authors assume that the Jacobian matrix is full-rank. The authors should give an example of a 2-layer (linear) NN to illustrate how this assumption is satisfied. I also wonder why the FIM is non-singular in this case since Eq 9 implies that the FIM is invertible.
>
> The condition, that the Jacobian is full rank means that in the “new parametrization space” our movement is not limited, the parametrization $\mathcal{P}(\mathbf{w_t})$ can change in all direction, which seems like a reasonable expectation.
> About the comment on equation 9: we can see from the calculation of Appendix A, that the Fisher information matrix is invertible in equation 9.
>
> >I wonder whether the authors consider the most common cases when the number of input features (D) << the number of (training) data points (N) << the number of NN parameters.
>
> The $D<N$ case was also a question of ours. Unfortunately, theoretically we were not able to give meaningful results, but experimentally we did consider this case as well. The result can be seen on figure 4 on the left hand side.
>
> > According to the caption of Figure 4, the number of training data points is 2500 and the number of input features is 1000. I do not think the empirical approximation of the FIM over 2500 training data points is good enough since the FIM should be at least a 1000-by-1000 matrix. I would like to see the performance of NGD when the number of input features is changed from 1000 to 50. For the sparse classification task, the authors may set the first 5 components to be 1 instead of the first 20 components.
>
> Thank you for the suggestion of additional experiments. We ran the experiment you suggested with 50 dimensions instead of 1000 and added it to the end of the paper (appendix figure 6). We ran an additional experiment that may address your original question more directly, in which we average the Fisher information over a much larger test set (approximating the population Fisher). The performance is indeed better, but still not as good as EGD, especially not as good as the diagonal parametrization. Please see the new dashed line in figure 4.
>
> >It will be great if the authors can comment on cases when damping is used. Does damping introduce extra inductive bias of NGD?
>
> Nice question. In some of our experiments (not reported here) we used small amounts of damping in order to solve numerical stability issues when matrices were ill-conditioned. Technically, yes, damping would introduce parametrization-dependent inductive biases. One can in fact interpolate between NGD and EGD by varying the amount of damping, and this interpolation is in fact studied in (Amari, 2020). It would be an interesting extension of our work to investigate how the behaviour changes as different levels of damping are added.
>
> >NGD is a discretization of NGF. Solving NGF is an initial value problem (IVP) in this setting. Thus, initialization should play a role such as pre-training. The statement about initialization should be more carefully stated.
>
> Thank you for pointing this out. We made changes to the article in section 5.2.” Q&A: Does initialization play a role?” to give a better explanation about the role of initialization.
>
> Thank you for all your nice insights and ideas. All of them were really useful and helped us in improving the article. We hope we could clear up some of the issues. We have also uploaded the updated article. If you have any more comments or suggestions please let us know.

---

### Official Review · Reviewer_VPSX · 2021-11-01

**Correctness:** 4
**Technical Novelty And Significance:** 2
**Empirical Novelty And Significance:** 1
**Recommendation:** 5
**Confidence:** 3

**Main Review:**

**Strengths:**
- I think studying NGD as "a form of ablation by eliminating parametrization-dependence" is a very clever way to better understand the complex interaction of the learning rule (and hyperparameters) and the parametrization of the model when it comes to explaining generalization performance in modern deep networks.
- The entire paper was very well organized, writing was clear and the proofs and theorem statements were well written and explained.
- I thought the background on the interaction of parameterization and EGD in separable classification and matrix completion was very well done.

**Weaknesses:**
 - The major weakness of this work is that while I think studying NGD (either theoretically or empirically) as a means of understanding the role of parametrization in generalization is very original, in the two settings studied (linearity separable classification and matrix factorization) the parameter-to-hypothesis mapping is already well understood.  I was hoping to see how this "ablation" technique could be applied to settings where there wasn't already a good understanding of the role of parametrization to provide *new* insights.  Clearly stating what new insights can be gained from this perspective or considering a not well studied setting (empirically or theoretically) would be helpful.
 - If the contributions of section 3 are to provide new, useful insight into the inductive bias of NGD, then I think this could be emphasized and discussed more.  What do these simplified settings imply for understanding NGD in a general setting?  I think you could expand here on the question in section 5, "Q: Does this mean NGD does not generalize well?".
 - You mention multiple times how you extend the work of Bernacchia et al. (2018) to develop "an algorithm that exploits the structure in the Fisher information matrix...for diagonal networks", but left all detail of this work in the Appendix.  Without diving into the details, its hard to understand what you did to extend their work.  This is one of your major contributions and certainly essential to your experimental section, so I would discuss this in much more detail in the body.

**Minor comments:**
- In the emphasized block, "influences the inductive biases of gradient-base learning", do you mean "gradient-based"?
- The end of this sentence is confusing: "...matrix factorization models tended to the minimum nuclear norm solution, but for fewer observed entries, which is the interesting case, this was not the case." Did you forget a word or meant to delete something?
- In the proof of "Invariance of NGF under reparametrization" you write $\mathcal{P}(\dot{w}_t)$ but I think you mean $\dot{\theta}_t$?  Correct me if I am wrong but in general these should only be the same if $\mathcal{P}$ is linear.
- In theorem 4 you write "the Jacobi matrix", do you mean "Jacobian"?
- Consider the relatively recent work "Kernel and Rich Regimes in Overparametrized Models" when discussing the role of initialization, parametrization and implicit biases.

**Summary Of The Paper:**

This paper studies the role of parameterization in determining inductive biases and influencing generalization in deep learning by studying natural gradient descent.  As the authors review, natural gradient descent is invariant to reparameterization.  This implies that the trajectory of NGD is in some sense ablating the role of parameterization allowing the authors to (1) determine the importance of parameterization with gradient descent in generalization, and (2) study the inductive biases unique to NGD.  The authors consider two settings both theoretically and empirically: separable classification with deep linear models and matrix completion via deep matrix factorization.

**Summary Of The Review:**

In summary, I think this work is very well written, clear, and addresses an important question (how to understand the role of parameterization in generalization) through an original lens, the study of natural gradient descent/flow.  However, the authors did not convince me that this perspective could lead to new insights that were not already well known.

---

> ### Author Response · Authors · 2021-11-18
> **Response to Reviewer VPSX**
>
> We greatly appreciate your insightful feedback. Our responses are provided below:
>
> >In the proof of "Invariance of NGF under reparametrization" you write $\mathcal{P}(\dot{ \mathbf{w_t}})$
>  but I think you mean $\dot \theta$
> ? Correct me if I am wrong but in general these should only be the same if $\mathcal{P}$ is linear.
>
> If we understood correctly, the confusion in the proof of  “Invariance of NGF under reparametrization” came from the unfortunate notation $\dot{(\mathcal{P}(\mathbf{w_t}))}=\frac{\partial \mathcal{P}(\mathbf{w_t}) }{\partial t}$ which equals $\dot \theta$ and what we tried to clarify. Importantly, the invariance holds for non-linear reparametrization as well.
>
> >You mention multiple times how you extend the work of Bernacchia et al. (2018) to develop "an algorithm that exploits the structure in the Fisher information matrix...for diagonal networks", but left all detail of this work in the Appendix. Without diving into the details, its hard to understand what you did to extend their work. This is one of your major contributions and certainly essential to your experimental section, so I would discuss this in much more detail in the body.
>
> We extended  (Bernacchia et al. 2018) in two ways: They considered only fully connected linear neural networks with a scalar output, we derive exact natural gradients for linear diagonal networks and deep matrix products. Secondly, we use more numerically stable algorithms for solving large linear systems involving Kronecker products (Fausett and Fulton (1994)  Large  Least  Squares  Problems  Involving  Kronecker Products) based on QR decomposition. These explanations can be found in the first paragraph of section 3.1 and the last but one paragraph in section 4. The remaining part is mostly computation, therefore we did not include any further detail due to space constraints.
>
> >Consider the relatively recent work "Kernel and Rich Regimes in Overparametrized Models" when discussing the role of initialization, parametrization and implicit biases.
>
> Thank you for the suggestion to highlight the role of initialisation in the context of the ‘Kernel and Rich Regimes’ paper - we have added this reference to the discussion section.
>
> > I was hoping to see how this "ablation" technique could be applied to settings where there wasn't already a good understanding of the role of parametrization to provide new insights.
>
> The idea behind our work was to choose problems which are well understood with the method GD, run NGD on them and compare our findings to the existing results in order to gain deeper understanding of the importance of parametrizations. However, it is a great suggestion and we will keep it in mind for future work.
>
> >If the contributions of section 3 are to provide new, useful insight into the inductive bias of NGD, then I think this could be emphasized and discussed more. What do these simplified settings imply for understanding NGD in a general setting? I think you could expand here on the question in section 5, "Q: Does this mean NGD does not generalize well?".
>
> This is certainly true, so we added some more discussion about this in Section 5 "Q: Does this mean NGD does not generalize well?" paragraph.
>
> Thank you for all the insightful comments and for bringing to our attention the typos and confusing sentences, we have made corrections. Your comments were really helpful and we hope we could resolve some of your issues. We have also uploaded the updated version of the article. If you have any other comments to add please contact us.

---

### Official Review · Reviewer_gWo5 · 2021-11-09

**Correctness:** 4
**Technical Novelty And Significance:** 3
**Empirical Novelty And Significance:** 3
**Recommendation:** 6
**Confidence:** 3

**Main Review:**

## Strenghts

 - The background is clearly explain. The illustrative figures, and the toy experiments are well designed.
 - The theorems are simple, yet as far as I know novel, and not trivial. The simple conclusions of these theorems provide useful insights into the inductive biases of natural gradient algorithms.

## Weaknesses

 - It is not clear from the text what is N and D in the beginning of section 3
 -  For theorem 1 to hold true, I think that $\beta_t$ and $\beta'_t$ also must start from the same initial conditions (it is not specified in the theorem statement)
 - I found some parts to be not very clear: between theorem 3 and 4 there is a discussion that starts with "OLS works very differently from large-margin methods" and end with "when $D>N \log N$ they end up finding the same solution".
 - section 4, paragraph 1: why is it a bad thing that "other entries won't move" ?

And on a different subject, I would have appreciated an experiment on an actual dataset/architecture to illustrate the consequence of your theorems in actual deep nets.

## Typos

 - section 3: conclusion missing $_t$ in $\beta_t$
 - "perfectly)" (second to last paragraph end of page 5)
 - page 6 "to unobserved entries"
 - "develope"

## Other related works that could have been discussed

- https://arxiv.org/abs/2006.10732
- https://arxiv.org/abs/2008.07545

**Summary Of The Paper:**

This paper studies the inductive bias of natural gradient flow in deep linear networks and in matrix factorization. They show that the solution to the empirical risk minimization problem using NGF is invariant to pixel permutation, and conclude that the desirable property of Euclidean gradient descent under logistic loss of finding the minimum l2 norm solution does not hold (theorem 1&2).

They also contribute an efficient NGD algorithm for deep diagonal linear networks.

**Summary Of The Review:**

Overall this is an interesting discussion of the different inductive biases of EGD and NGF.

---

> ### Author Response · Authors · 2021-11-18
> **Response to Reviewer gWo5**
>
> Thank you for your useful comments. In reply to your concerns:
>
> >It is not clear from the text what is N and D in the beginning of section 3
>
> We made some clarifications about the meaning of $N$ and $D$ in section 3 where $N$ denotes the number of data points and $D$ denotes the number of input features.
>
> >For theorem 1 to hold true, I think that $\beta_t$ and $\beta_t’$ also must start from the same initial conditions (it is not specified in the theorem statement)
>
> Thank you for pointing this out, it is indeed true that Theorem 1&2 only hold if $\beta_t$ and $\beta_t’$ start from the same initial conditions, which was included in the proofs but now the statements have been corrected as well.
>
> >section 4, paragraph 1: why is it a bad thing that "other entries won't move" ?
>
> In case of matrix completion if we assume that the Jacobian is full rank, NGD finds the same solution as EGD with direct parameterization ($\beta = W$), namely where we have observations the entries converge to the observations and the others don’t move. This behaviour is not beneficial, because it fails to generalize, which is the aim of matrix completion tasks.
>
> >I found some parts to be not very clear: between theorem 3 and 4 there is a discussion that starts with "OLS works very differently from large-margin methods" and end with "when $D>N\log N$ they end up finding the same solution".
>
> We are sorry for being unclear about this, we have made changes in the article and we are also including some clarification here.
> In general cases, OLS interpolation and large-margin (LM) methods find qualitatively different solutions in classification tasks. While the LM solution is typically a linear combination of a small subset of training data (the support vectors), in OLS all datapoints are support vectors. As shown in (https://arxiv.org/abs/2009.10670v1), under some conditions this difference disappears in the highly overparameterized regime - when $D>N\log N$. An implication of Theorem 3 is that this phenomenon, known as support vector proliferation, occurs in NGF when $D>N$. Thus there is a regime where NGF and EGF find qualitatively different classifiers, with different generalisation properties (https://arxiv.org/abs/2009.10670v1).
>
> >And on a different subject, I would have appreciated an experiment on an actual dataset/architecture to illustrate the consequence of your theorems in actual deep nets.
>
> Although it would be interesting to run some experiments on actual dataset/architecture, however, this would require substantially different implementation and high computational cost, we consider this beyond the scope of our current paper.
>
> We thank you for the excellent references which we now discuss in the related works section of the discussion. (Amari et al. 2020) is particularly closely related indeed, with the main difference being that they study squared loss, This makes the Fisher information matrix as well as NGD dynamics considerably simpler to our case under logistic loss. For example, the norm $||.||_P$ which appears in their Remark on page 4 is not even a norm in the case of NGF under logistic loss, and similar results do not hold in our case.
>
> Thank you for your useful insights. We hope we were able to resolve some of your concerns. We also uploaded the updated article. If in the remaining time you have any other comments or issues we might be able to work on please let us know.

---

### Official Review · Reviewer_2Yhk · 2021-11-09

**Correctness:** 3
**Technical Novelty And Significance:** 3
**Empirical Novelty And Significance:** 2
**Recommendation:** 5
**Confidence:** 2

**Main Review:**

The main subject of the paper is to utilize properties in NGD and study the inductive bias of NGD. However, either current reviewer or the authors are misunderstanding what the “reparametrization invariance” properties of NGD means. In general, the descriptions in section 2.3 where the NGD is described looks okay. The authors even say that

“NGD with infinitesimally small learning rate (i. e. NGF) always follows the same trajectory in model-space and this finds the same optimum, irrespective of how it is parametrized, provided that the parametrization is _smooth_ and _locally invertible_.”

However, as far as I can tell, the authors study parameterization changes that’s not considered reparameterization. For example, in Section 3, the authors consider $\beta= w$ vs $\beta = w_1 \odot \cdots \odot w_L$, the parameter space dimension changes from $D$ to $L \times D$ which to me is not considered reparameterization with guaranteed optimization equivalence under NGD. Similarly in matrix completion Section 4, parameterizing $\beta$  (with dimension $D \times D$) with different depth of matrices $W_i$ (dimension $L \times D \times D$) to me is just a completely different model architecture rather than a reparameterization.  To me, what others are saying is similar to  ResNet-50, ResNet-101 or VisionTransformer to solve ImageNet classification problems are “different parameterization” in reality are just different architectures.

Without the justification of connecting optimization of different architectures, I don’t quite understand the point of the analysis. There’s a chance that the reviewer is totally missing the point but to say the least, the main point, if it exists, doesn’t come across properly given that NGD doesn’t say anything about optimization of different architectures.

To delve into this issue in more detail, for example in Figure 5 / Section 4, where the authors identify a problem where NGD doesn’t generalize but GD does well with L=2, L=3. I do wonder what the author is actually showing is that L=1 is a bad way (in an obvious way) to solve matrix completion problems. Can authors confirm the NGD was also run on L=2, L=3 cases and obtain non-generalization behavior?

The reviewer does recognize in Figure 2 that C, D leads to the same solution which is interesting and would like to have better understanding given that NGD doesn’t guarantee the same optimization for different architectures.


Nit:
- NGF is used without introduction: since the paper mainly focuses on NGF, one should introduce and define early in the introduction.
- p3 Figure 3 description needs more explanation, without reading section 4, it’s hard to tell what the figure is showing. At least describe that they are components of a natural NTK.
- p5 (X, y) -> $(X, y)$ (put in the math-form)
- p8  “from observations too unobserved entries” -> “from observations to unobserved entries”
- p8 “ichi Amari”: remove “ichi”
- p14 “Until know this” -> “Until now this”

-----------------------------------------------------
Post-rebuttal: Thank you for the detailed explanation on validity of reparametrization invariance and verifying NGD in deep MF models. I agree that deep linear models the paper study is somewhat special. Also intermediate layer width (e.g. extreme bottleneck width) could break local invertible map, although the paper simplifies the analysis to equal large hidden layer widths. It would be probably worth pointing out and explaining how reparametrization invariance can be applied to models under study. Also the author should justify full rank of FIM.

I've raised my score and lowered my confidence based on the discussion.


**Summary Of The Paper:**

The paper studies inductive bias of natural gradient descent (NGD) in deep linear networks. Utilizing _reparameterization invariance_ properties of NGD, the authors attempt to eliminate the role of parameterization and isolate the effect of solution found by gradient descent.  The paper studies gradient flow dynamics for 1) separable classification under logistic loss and 2) deep matrix factorization. One of the contributions stated is identifying learning problems where NGD fails to generalize while GD with the right architecture performs well.

**Summary Of The Review:**

While studying inductive bias of gradient descent optimization and their role to generalization property is an important subject and natural gradient descent provides an interesting tool to investigate and isolate certain optimization artifacts, I believe the current paper’s direction of analyzing these may be flawed. Without proper justification of their methodology, the reviewer suggest that the paper shouldn’t be accepted to ICLR.

---

> ### Author Response · Authors · 2021-11-12
> **Clarification on different architectures**
>
> Thank you for the review. We hope the clarification below will help resolve the confusion, and will allow you to reevaluate the work, while we are making improvements to the manuscript.
>
> ### Conditions for the equivalence to hold
>
> For NGD to be invariant across two architectures/parametrisations, several conditions have to hold. For example, if (a) the two architectures map to the same hypothesis space, (b) this common hypothesis space forms a smooth p-manifold, and (c) the parameter-to-hypothesis map is smooth with rank-p Jacobian at all points along the optimisation trajectory, one can say that NGF in the two architectures will be equivalent in the sense we mean in the paper.
>
> These conditions all hold between the deep linear architectures we study, irrespective of depth (barring some pathological points we never encounter). We can see now how the phrasing _smooth and locally invertible_ was misleading. In reality, when the parametrisations have different dimensionality, we require the mapping to be smooth and locally invertible **within a p-dimensional subspace around each point**, where p is the dimensionality of the common hypothesis space as above.
>
> To be very clear we DO NOT EXPECT the invariance to hold when comparing wildly different neural network architectures such as a ResNet to a Transformer, as conditions (a) and (b) above most likely do not hold. But we also want to clarify that this is not a claim we make in the paper, and we will clarify this in the text to avoid confusion.
>
> In summary, we can study invariance of NGF across different architectures because these are all deep linear models, so they are all smooth parametrizations of the same hypothesis space.
>
> ### How we used these insights
>
> However, we do not rely on this strong notion of invariance across architectures for the main arguments of our paper. Let us reframe our contribution in this light.
>
> We wish to better understand contributing factors to good generalisation in deep learning. One potential contributing factor is the parameter-to-hypothesis map (the structured, layered way we parametrise smooth functions in deep learning). This map can influence generalisation through multiple mechanisms:
>
> 1. Initialization: Initialization of deep networks is trivial in parameter-space, but the nontrivial parameter-to-hypothesis mapping may give rise to non-trivial priors/initial conditions in hypothesis space. For example, [Valle Pérez et al (2019)](https://arxiv.org/abs/1805.08522) observe that Gaussian initialization induces simplicity bias (measured by e.g. Kolmogorov-complexity) at initialization, and that this is true in several commonly used architectures.
> 2. Influencing dynamics and GD trajectories: As shown by the works referenced in the background section, the parameter-to-hypothesis mapping influences the dynamics of gradient descent, biasing it towards finding minima which have low l2 norm in parameter space. l2 norm is a trivial regulariser, but when mapped by a non-trivial parameter-to-hypothesis map, it can induce non-trivial implicit regularisation in hypothesis space, such as towards sparse or low-rank solutions. It is speculated that this bias of GD in deep architectures (e.g. ResNet) gives rise to some form of useful implicit regularisation, though one which is not well understood.
> 3. Influencing the structure of the hypothesis space: Finally, the parameter-to-hypothesis mapping also determines what hypotheses are even implementable by the specific architecture.
>
> What our work says is this: within a specific architecture, replacing EGD with NGD but keeping everything else the same essentially disturbs the second mechanism above. Any implicit regularisation effect, whatever it is, that the specific architecture confers through influencing GD trajectories should disappear. The inductive biases would instead be replaced by inductive biases of natural gradients, which only depend on the information geometry of the hypothesis space.
>
> Thus, NGD can be used as a tool to create informative ablation scenarios where one of the possible contributing factors to generalisation are intervened on, while the influence of others remain. We suggest that such experiments should therefore be useful in our pursuit to disentangle sources of implicit regularisation in deep learning and to understand their relative practical importance.
>
> In this paper we demonstrate the usefulness of this NGD-as-ablation approach by studying architectures where generalisation advantage can be attributed entirely to mechanism 2 above, and where the behaviour of NGD can be predicted analytically. We indeed find that in these situations, the ‘NGD-as-ablation’ technique works, inasmuch as generalisation indeed suffers when EGD is replaced by NGD.
>
> ### NGD in deeper MF models
>
> To your specific question, we can confirm that we ran NGD for different depth matrix factorisation models, and they fail to learn irrespective of depth, step size or initialization scale.

---

### Official Review · Reviewer_CBDn · 2021-11-10

**Correctness:** 3
**Technical Novelty And Significance:** 2
**Empirical Novelty And Significance:** 2
**Recommendation:** 5
**Confidence:** 3

**Main Review:**

This paper provides some interesting analyses on natural gradient descent in training deep linear models. The observation that natural gradient flow is invariant to reparameterization looks particularly insightful. Experient results are also provided to back up the theories.

However, I also find several limitations of the paper:

- The presentation of the paper is not clear. For example, the studies in this paper are focused on binary classification with logistic loss. However, Section 2.2 and Section 4 discussed matrix factorization, which, if my understanding is correct, is a regression-type problem. Therefore, it is not clear how the analyses in this paper can be applied to matrix factorization problems. Moreover, the remark below Theorem 3 is not consistent with the introduction around equation (5).

- The results of this paper are not very comprehensive. Theorems 3 and 4 seem to be much weaker compared with the inductive bias results for Euclidean gradient flow. Moreover, the claim that there exist learning problems where natural gradient descent perform much worse than Euclidean gradient descent is only demonstrated experimentally.


**Summary Of The Paper:**

This paper studies the inductive bias of natural gradient flow. The authors highlight the invariance property of natural gradient flow under reparameterization, and study the training dynamics of natural gradient flow under different problem settings for deep linear models.

**Summary Of The Review:**

For the reasons listed in the previous section, I think this paper is at the borderline, and I currently tend to recommend rejection.

---

> ### Author Response · Authors · 2021-11-18
> **Response to Reviewer CBDn**
>
> Thank you for your effort in reviewing our paper. Below can be found our responses to the concerns you raised.
>
> >The presentation of the paper is not clear. For example, the studies in this paper are focused on binary classification with logistic loss. However, Section 2.2 and Section 4 discussed matrix factorization, which, if my understanding is correct, is a regression-type problem. Therefore, it is not clear how the analyses in this paper can be applied to matrix factorization problems.
>
> In our paper we investigated the effects of Natural gradient descent compared to Euclidean gradient descent. In the case of linear classification we found some interesting behavior (Theorem 3&4), however there exist other learning problems where NDG fails to generalize, like matrix completion. Our results for linear classification do not apply for matrix completion.
>
> >Moreover, the remark below Theorem 3 is not consistent with the introduction around equation (5).
>
> Thank you for pointing this out, we made changes in the remark below Theorem 3 to make it more consistent with the definition of NGF (given in the paper) The fact that there are several possible paths for NGF (as described around equation 5) comes from $F(\beta)$ being singular and with the choice of $s=X\beta $ we give a description of the NGF  trajectory in the eigenspace of $F(\beta)$.
>
> >Moreover, the claim that there exist learning problems where natural gradient descent perform much worse than Euclidean gradient descent is only demonstrated experimentally.
>
> We added a statement & proof to section 4 which demonstrates theoretically that NGD performs worse in the case of matrix completion, than EGD with the right parametrization.
>
> Thank you for your useful comments, we have uploaded the updated article. Let us know if there is anything else we could improve on in the remaining time.

---

### Decision · Program_Chairs · 2022-01-20

**Decision:**

Reject

**Comment:**

*Summary:* Study inductive bias of natural gradient flow.

*Strengths:*
- Some reviewers found the invariance to reparametrization insightful, a good way to better understand the interaction of the learning rule and parametrization.
- Experiments support the theory.

*Weaknesses:*
- Unclear takeaway message.
- Comparison with Euclidean case not comprehensive.
- Insufficient distinction between reparametrization (invertible) and different parametrization. No experiments on actual dataset/architecture.
- Some reviewers found the the cases considered in the paper are already well understood.

*Discussion:*

2Yhk found that although the author responses and other reviews clarified some of their concerns, particularly about reparametrization conditions, the result provided in the paper is not strong enough and could be further clarified. The authors found that this reviewer might have misunderstood the paper. Following the discussion period, the reviewer raised his/her score and lowered his/her confidence. In response to CBDn the authors added demonstration of NGD being worse than EGD on matrix completion. In one of the responses, the authors summarize their contribution as: ''replacing EGD with NGD … disturbs the second mechanism [dynamics and GD trajectories]''. I find the question really is what kind of quantitative conclusions can be made. gWo5 pointed out important related work that was not discussed in the initial submission. Authors added discussion. VPSX misses applications to less well understood settings. Authors however only offer to keep this in mind for future work. VPSX also asks to emphasize the insights into the inductive bias of NGD. Authors added some discussion, however mostly pertaining previous works and not specific enough in my opinion.

*Conclusion:*

One reviewer found this work marginally above the acceptance threshold and four other reviewers found that it does not reach the bar for acceptance. I find the topic worthwhile and that it deserves a thorough investigation. However, I concur with the reviewers that some concepts require a clearer presentation and that it would be desirable to see more general results and more comprehensive discussions. Several suggestions were made by the reviewers and acknowledged by the authors, but many of these were left for future work. In summary, I think that the article makes a good start but needs more work. Therefore I am recommending a rejection at this time. I encourage the authors to revise and resubmit.